# Eya2 promotes cell cycle progression by regulating DNA damage response during vertebrate limb regeneration

**Konstantinos Sousounis[1,2], Donald M Bryant[1], Jose Martinez Fernandez[1], Samuel S Eddy[3], Stephanie L Tsai[1,4], Gregory C Gundberg[1,2], Jihee Han[1], Katharine Courtemanche[1], Michael Levin[2,5], Jessica L Whited[1,2,6,7]***

[1]Department of Stem Cell and Regenerative Biology, Harvard University, Cambridge, United States; [2]The Allen Discovery Center at Tufts University, Medford, United States; [3]Department of Orthopedic Surgery, Boston, United States; [4]Department of Molecular and Cellular Biology, Harvard University, Cambridge, United States; [5]Department of Biology, Tufts University, Medford, United States; [6]The Harvard Stem Cell Institute, Cambridge, United States; [7]The Broad Institute of MIT and Harvard, Cambridge, United States

**Abstract** How salamanders accomplish progenitor cell proliferation while faithfully maintaining genomic integrity and regenerative potential remains elusive. Here we found an innate DNA damage response mechanism that is evident during blastema proliferation (early- to late-bud) and studied its role during tissue regeneration by ablating the function of one of its components, Eyes absent 2. In *eya2* mutant axolotls, we found that DNA damage signaling through the H2AX histone variant was deregulated, especially within the proliferating progenitors during limb regeneration. Ultimately, cell cycle progression was impaired at the G1/S and G2/M transitions and regeneration rate was reduced. Similar data were acquired using acute pharmacological inhibition of the Eya2 phosphatase activity and the DNA damage checkpoint kinases Chk1 and Chk2 in wild-type axolotls. Together, our data indicate that highly-regenerative animals employ a robust DNA damage response pathway which involves regulation of H2AX phosphorylation via Eya2 to facilitate proper cell cycle progression upon injury.

**\*For correspondence:**
jessica_whited@harvard.edu

**Competing interests:** The authors declare that no competing interests exist.

## Introduction

DNA damage response (DDR) is an important process in ensuring cellular function. Genotoxic stress can cause senescence and apoptosis in proliferating cells or carcinogenesis when oncogenes are affected (*Zeman and Cimprich, 2014*; *Vitale et al., 2017*). Upon DNA damage, sensor proteins initiate a DDR that eventually guides DNA repair enzymes to resolve the lesion (*Huen and Chen, 2008*). The histone variant H2AX is considered a major DNA damage mediator protein because it is used as a scaffold for protein-protein interactions based on its amino acid modifications. H2AX phosphorylated at the 139 serine, commonly known as gamma-H2AX (γ-H2AX), is widely used as a DNA damage marker because it is found aggregated near lesions shortly after they occur (*Huen and Chen, 2008*; *Yuan et al., 2010*; *Rogakou et al., 1998*). Acting in concert to ensure genomic stability, cell cycle checkpoints ordinarily prevent cells from progressing before resolving DNA damage. Cell cycle arrest through activation of DNA damage checkpoints via Chk1 and Chk2 is an important part of the DDR mechanism (*Zhou and Elledge, 2000*; *Bartek and Lukas, 2003*; *Smith et al., 2010*). DNA repair resolves the DDR and protein modifications are reversed so normal cellular function can be resumed (*Shaltiel et al., 2015*).

However, DDR does not always lead to DNA repair; molecular safeguards mediate critical decisions as to whether the amount of DNA damage is amendable, or whether apoptosis should be triggered instead (*Cook et al., 2009*). These molecular switches, such as p53, are known tumor suppressors and, when deregulated, apoptosis is circumvented, and carcinogenesis could occur (*Lakin and Jackson, 1999*). For instance, in vitro cultured human stem cells experience DNA damage and chromosomal rearrangements that change their cellular fate and render them carcinogenic and unsuitable for use in clinical applications in regenerative medicine (*Vitale et al., 2017*). On the other hand, amphibians can faithfully regenerate body parts throughout life, often even when repeatedly challenged (*Bryant et al., 2017a*; *Bryant et al., 2017b*; *Sousounis et al., 2015*; *Eguchi et al., 2011*). Replicative stress and reactive oxygen species generated during the process could challenge regeneration, at the cellular level by inhibiting cell cycle progression or at the functional level by losing the ability to form a proper structure (*Sousounis et al., 2014*; *Al Haj Baddar et al., 2018*; *Ferreira et al., 2016*).

How do amphibians regulate genotoxic stress and DDR while ensuring tissue regeneration? Previous studies have shown that during early phases of regeneration, the activity of p53 is reduced to allow for progenitor cell cycle re-entry (*Yun et al., 2013*), but it is later activated to enable differentiation (*Villiard et al., 2007*). Since p53 is a major regulator of DDR, amphibians might be de-sensitized to DNA damage allowing for rapid progression through the cell cycle checkpoints. Their massive genome could also facilitate this process; it has been previously shown that DNA damage is more likely to occur in areas that do not encompass coding regions and regulatory elements, like retrotransposons, which are significantly abundant in amphibian genomes (*Poetsch et al., 2018*; *Smith et al., 2019*; *Nowoshilow et al., 2018*; *Elewa et al., 2017*). A completely opposite (but not mutually exclusive) strategy would entail the initiation of a DDR together with the onset of the regeneration process to ensure rapid DNA repair and cell cycle progression. Studies that have repeatedly initiated the regeneration process, for example when newt lenses were removed and successfully regenerated 19 times over 18 years, support this hypothesis since the structure and function of the genome would remain intact over many rounds of cell division (*Sousounis et al., 2015*; *Eguchi et al., 2011*; *Sousounis et al., 2014a*). In addition, high-throughput gene expression datasets acquired from regenerative tissues show up-regulation of DNA repair genes across multiple organs and animal species, indicating that a proper DDR might be an innate part of the regenerative mechanism (*Sousounis et al., 2014*; *Darnet et al., 2019*; *Barghouth et al., 2019*; *Shiroor et al., 2019*; *Bedelbaeva et al., 2010*). This mechanism could occur in conjunction with clearing senescent cells from regenerating tissues, which would allow only resilient proliferating progenitors to be present in the blastema (*Yun et al., 2015*).

Here, we address these two hypotheses in the context of axolotl limb regeneration. We first investigated the relationship between DDR and tissue regeneration. We found an elevated DDR following limb amputation, which was accompanied by reduced relative γ-H2AX levels and baseline amounts of DNA damage. Then we asked whether DDR is necessary for limb regeneration. To answer this question, we genetically targeted a DDR- and H2AX-associated gene, *eyes absent 2* (*eya2*). Eya2 regulates H2AX phosphorylation status, mediates decisions regarding DNA repair (*Cook et al., 2009*), and has been shown to be up-regulated during regenerative processes (*Stewart et al., 2013*; *Sousounis et al., 2014b*). We generated Eya2 mutant axolotls and found that their regenerative capability was significantly reduced and their γ-H2AX levels were significantly elevated, especially in cells that have entered the cell cycle. Interestingly, we did not detect accumulation of DNA damage, indicating that our model specifically de-regulated the DDR process. RNAseq analysis of *eya2*$^{-/-}$ tissues and pharmacological inhibition of Eya2's phosphatase activity were also used to corroborate our findings. In addition, we uncovered parallel evidence that DDR is required for successful regeneration as inhibiting a separate component of this process, the proper control of the DNA damage checkpoints by Chk1 and Chk2, led to severe regenerative impairment. This study reveals that DDR is an innate part of the regeneration process and identifies Eya2 as a regulator of DDR and cell cycle progression during progenitor cell proliferation. How amphibians regulate DDR and allow for cellular proliferation while maintaining cell fate could reveal important clues for improving the use of human stem cells in regenerative medicine.

## Results

### Axolotl limb regeneration is associated with elevated DNA damage response and low genotoxic stress in progenitor cells

To identify the presence of a DNA damage response (DDR) following limb amputation, we analyzed transcriptomic profiles from previously published expression datasets acquired by RNAseq and microarrays that included a complete time course of regeneration (*Stewart et al., 2013*; *Voss et al., 2015*). We analyzed 190 genes associated with DDR and DNA repair from both datasets (*Supplementary file 1*). We found expression of DDR-related genes to be mostly up-regulated during regeneration (*Figure 1a*, Green). Genes also appeared up-regulated irrespective of their molecular role during DDR (*Figure 1—figure supplement 1a*). We validated the expression of 17 genes using qPCR and found that 11 (65%) of them were up-regulated during regeneration (*Figure 1—figure supplement 2*). We then focused on genes that have tentative functions in promoting DNA repair, as these genes may regulate DDR for proper cell cycle entry and regeneration (see Materials and methods, *Supplementary file 1*). The genes in this subcategory also appeared up-regulated during regeneration (+DDR, *Figure 1b*). Overall, our transcriptomic and qPCR analysis of DNA repair/damage genes revealed that DDR was evident as early as the early-bud blastema (~5 dpa) and peaked at the late-bud blastema (~14 dpa). To associate this response with the levels of actual DNA damage, we performed alkaline single cell gel electrophoresis (comet assay) and found that there are no differences between intact and 14 dpa blastema cells, whereas cells treated with UV showed more DNA damage (*Figure 1c* and *Figure 1—figure supplement 1b*). To corroborate these data, we focused on the DNA damage mediator histone H2AX. First, we analyzed the axolotl *h2afx* sequence using two strategies: phylogenetic tree construction (*Figure 1d*) and multiple sequence alignment (*Figure 1—figure supplement 1c*). We found that axolotl H2AX has a conserved carboxyl tail containing the S139, also known as $\gamma$-H2AX when phosphorylated, and Y142 (*Cook et al., 2009*; *Yuan et al., 2010*; *Figure 1—figure supplement 1c*, green box). We validated antibodies detecting these phosphorylation sites in vitro using AL1 axolotl fibroblast cells following UV treatment (*Figure 1—figure supplement 1d–e*). RNAseq and qPCR also revealed that *h2afx* expression was significantly up-regulated during regeneration (*Figure 1b* and *Figure 1e*). We then used western blot to determine the phosphorylation state of H2AX in axolotl tissues. Using samples acquired across different time points during limb regeneration, we found that the total amount of H2AX and pY142-H2AX were elevated compared to GAPDH starting at 5 dpa, corroborating the presence of a DDR in the early- and late-bud blastema (Materials and methods, *Figure 1—figure supplement 3*). To analyze this process in greater depth, we focused on blastema samples at 14 dpa, the time that DDR was found to be the most elevated (*Figure 1f*). We again found that the total amount of H2AX protein was increased during regeneration (*Figure 1g*), following the same trend as its mRNA levels (*h2afx*, *Figure 1b* and *Figure 1e*). The total levels of $\gamma$-H2AX protein were also increased (*Figure 1h*); however, the relative ratio of $\gamma$-H2AX to total H2AX protein was decreased during regeneration (*Figure 1i*). Overall, these data indicated that blastema cells may require more $\gamma$-H2AX to rapidly correct replication errors during proliferation but kept a low $\gamma$-H2AX/H2AX ratio to reduce potential genotoxic stress signals that could hinder cell cycle progression (*Rogakou et al., 1998*; *Fernandez-Capetillo et al., 2004*). Interestingly, we identified that the relative pY142-H2AX levels were increased at 14 dpa (*Figure 1j* and *Figure 1k*), which suggested that H2AX regulation was important during regeneration and a potential target to study in association with DDR and genome integrity. Together, these results suggest that DDR is evident during early-bud blastema and peaks at late-bud blastema, thereby preventing accumulation of DNA damage during regeneration. This may enable high levels of cellular proliferation despite local environmental genotoxic stressors. To further test this hypothesis, we aimed to genetically ablate one of the DDR components and identify whether challenging this process would have an impact on the regenerative capacity.

### Eya2 is widely expressed in early-bud blastema and is associated with DNA repair and cell cycle during regeneration

Since pY142-H2AX levels were found to be elevated during regeneration (*Figure 1k* and *Figure 1—figure supplement 3*), we focused on the Eyes absent (Eya) protein family, which are known

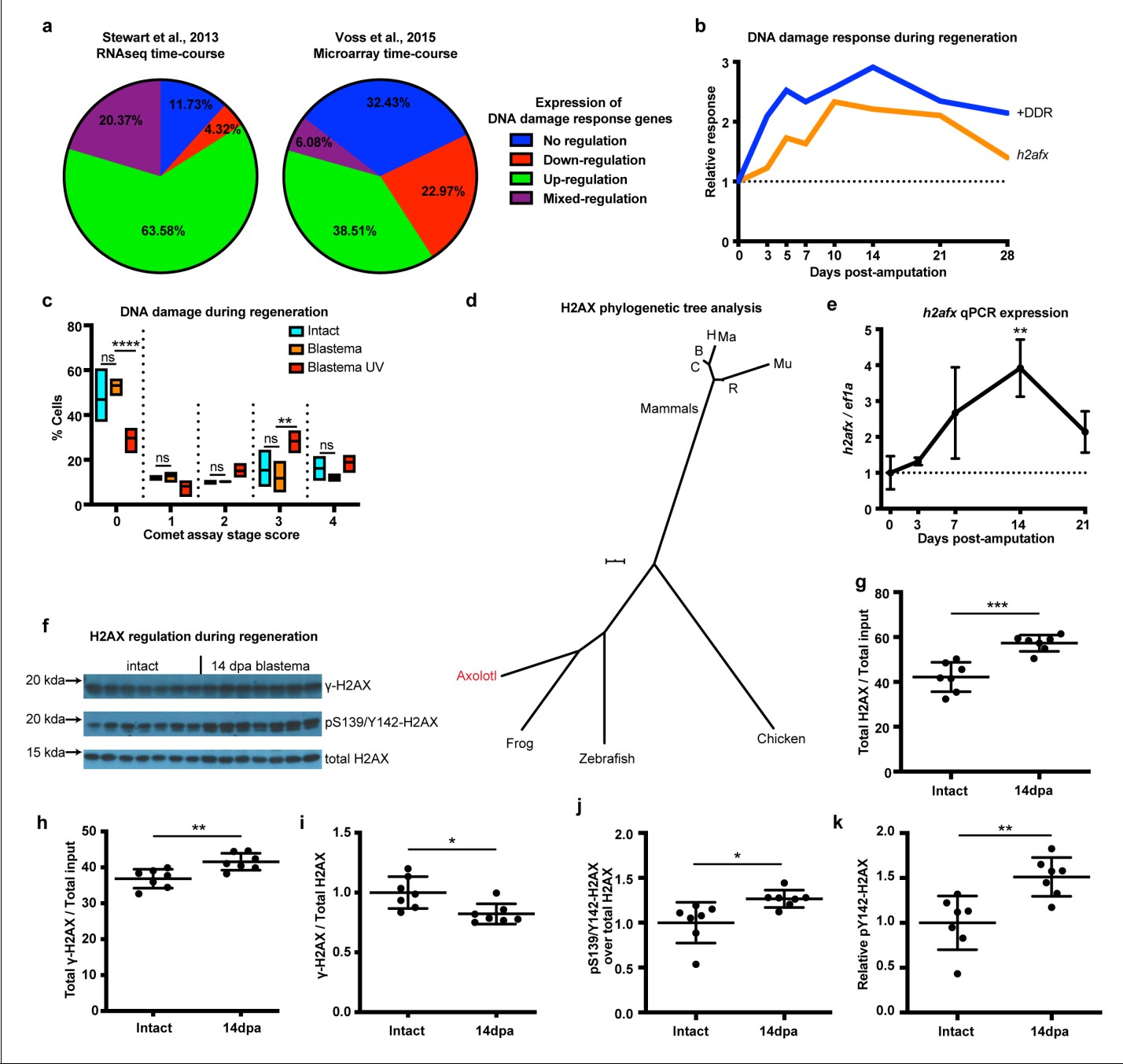

**Figure 1.** DNA damage response is elevated during limb regeneration and is associated with regulation of H2AX phosphorylation status. (a) Regulation of DNA damage response (DDR) genes in two high-throughput gene expression datasets. The majority of these genes appeared up-regulated during limb regeneration (green area). See Materials and methods for regulation categories. (b) Relative gene expression changes of DNA damage response genes (+DDR) and *h2afx* during regeneration compared to intact. (c) Categories represent percent of cells in each comet assay stage score with stage 0 cells having no detectable DNA damage and stage four having the most. There is no statistical difference between intact (cyan) and blastema (orange) cells for the presence of DNA damage. Blastema cells treated with UV prior to comet assay were used as control and depict statistical significance compared to untreated blastema cells at stage 0 with less percent of cells (Two-way ANOVA with Bonferroni's multiple comparisons test: p-value<0.0001) and at stage three with more percent of cells (p-value<0.01). The data indicates that there is no DNA damage accumulation during regeneration. (d) Phylogenetic tree analysis of axolotl H2AX sequence with other vertebrates. C: Canis, B: Bos, H: Human, Mu: Mus, Ma: Macaca, R: Rattus. Scale Bar: 0.01. (e) Validation of axolotl *h2afx* expression during limb regeneration using qPCR. (f) Western blot for H2AX and its phosphorylation status during regeneration. Quantification on (g, h, i, j, k). (g) Total H2AX levels in intact and 14 dpa blastema tissue normalized by the total protein input. There is up-regulation of H2AX total protein during regeneration (Unpaired t test, p-value<0.001). (h) Total γ-H2AX levels in intact

*Figure 1 continued on next page*

*Figure 1 continued*

and 14 dpa blastema tissue normalized by the total protein input. There is up-regulation of γ-H2AX total protein during regeneration (Unpaired t test, p-value<0.01). (i) Relative γ-H2AX levels in intact and 14 dpa blastema tissue normalized by the respective H2AX levels. There is down-regulation of relative γ-H2AX levels during regeneration (Unpaired t test, p-value<0.05). (j) Relative pS139/Y142-H2AX levels in intact and 14 dpa blastema tissue normalized by the respective H2AX levels. There is up-regulation of relative pS139/Y142-H2AX levels during regeneration (Unpaired t test, p-value<0.05). (k) Relative pY142-H2AX levels in intact and 14 dpa blastema tissue normalized by the respective H2AX levels. There is up-regulation of relative pY142-H2AX levels during regeneration (Unpaired t test, p-value<0.01). These data indicate that there is dynamic regulation of H2AX phosphorylation status during regeneration. ns denotes p-value>0.05, * denotes p-value<0.05, ** denotes p-value<0.01, *** denotes p-value<0.001, **** denotes p-value<0.0001. Vertical bars on plots represent standard error of the mean (SEM).

The online version of this article includes the following figure supplement(s) for figure 1:

**Figure supplement 1.** DDR pathways, comet assay staging, H2AX multiple sequence alignment and validation of H2AX antibodies.

**Figure supplement 2.** Validation of DDR gene expression profiles using qPCR.

**Figure supplement 3.** H2AX phosphorylation status at different timepoints during limb regeneration.

---

regulators of H2AX phosphorylation status and initiators of DNA repair through their phosphatase domain (*Cook et al., 2009*). We first investigated the spatial and temporal expression of individual members upon limb amputation. We found that *eya2* is the most highly-expressed gene of its family and is also highly up-regulated post-amputation (*Figure 2a*, *eya2*) (*Stewart et al., 2013*). We additionally performed qPCR and verified its expression profile during regeneration (*Figure 2b*). Phylogenetic tree analysis using the entire Eya family further validated homology to other vertebrate orthologs of Eya2 (*Figure 2c*). In order to associate Eya2 with DNA repair during limb regeneration, we performed a co-expression correlation analysis using RNAseq transcriptomic data from limb/blastema samples at 0, 3, 5, 7, 10, 14, 21, and 28 days following amputation (*Supplementary file 2*). Genes positively associated ($R^2$ >0.8, slope >0) and negatively associated ($R^2$ >0.8, slope <0) with *eya2* expression were used for Gene Ontology Enrichment Analysis (*Figure 2—figure supplement 1a and b*, respectively). Using this method, we found that the expression profiles of genes involved in DNA repair and cell cycle were positively correlated with *eya2* expression, suggesting Eya2's involvement in these processes during limb regeneration (*Figure 2d*, blue bar). Conversely, mature muscle genes were negatively correlated with *eya2* expression, consistent with *eya2*'s role during early muscle development in other vertebrates (*Heanue et al., 1999*; *Figure 2d*, red bar). To test whether the axolotl Eya2 protein physically interacts with an H2AX-containing complex in cells, we performed an immunoprecipitation experiment using a tagged version of the axolotl Eya2, and we detected the phosphorylated form of H2AX in the precipitate (*Figure 2e*). Taken together, these data suggest that Eya2 plays a role in cell cycle and DNA repair regulation during regeneration and that Eya2 interacts with phosphorylated H2AX.

To identify which cells express *eya2* during regeneration, we performed RNA in situ hybridization and found *eya2* mRNA detectable in the majority of blastema cells and wound epidermis at 5 dpa (red puncta, *Figure 2f*). These cells are known to encompass limb progenitors of all lineages including skin, muscle, connective tissue, and cartilage (*Whited et al., 2013*; *Currie et al., 2016*; *Gerber et al., 2018*; *Leigh et al., 2018*). Indeed, time-course double in situ hybridization of *eya2* and *prrx1*, a connective tissue progenitor marker (*Gerber et al., 2018*), further validated *eya2* expression in the connective tissue lineage. We found that *eya2* and *prrx1* co-localized in connective tissue progenitors throughout regeneration, and we further observed *eya2* expression in muscle as well. In intact tissue, *eya2* and *prrx1* were present in periosteal cells, which are known to participate in cartilage regeneration (*McCusker et al., 2016*), while *eya2* alone was present in the muscle (*Figure 2g and h*). In the early-bud blastema (7 dpa), *eya2* and *prrx1* showed expression throughout the blastema with *prrx1* being very robust (*Figure 2g'*). In the mid- to late-bud blastema (14 dpa), *eya2* and *prrx1* started to compartmentalize, with *eya2* having strong expression in the developing muscle which appeared to be contiguous with the mature fibers in the stump, and *prrx1* aggregating in the rest of the blastema (*Figure 2g''*). Interestingly, *eya2* and *prrx1* double-positive cells were clearly present in the boundary between high-*eya2* and high-*prrx1* expressing areas (*Figure 2h'*). This suggested that these areas contained early connective tissue progenitor cells that still expressed *eya2* and, as these cells mature, they become *prrx1*-only positive in the center of chondrogenesis (*Balic et al., 2009*). This phenomenon was more evident in palette stage limbs (21 dpa) where muscle/cartilage boundaries were more defined (*Figure 2g''' and h''*). We also identified that

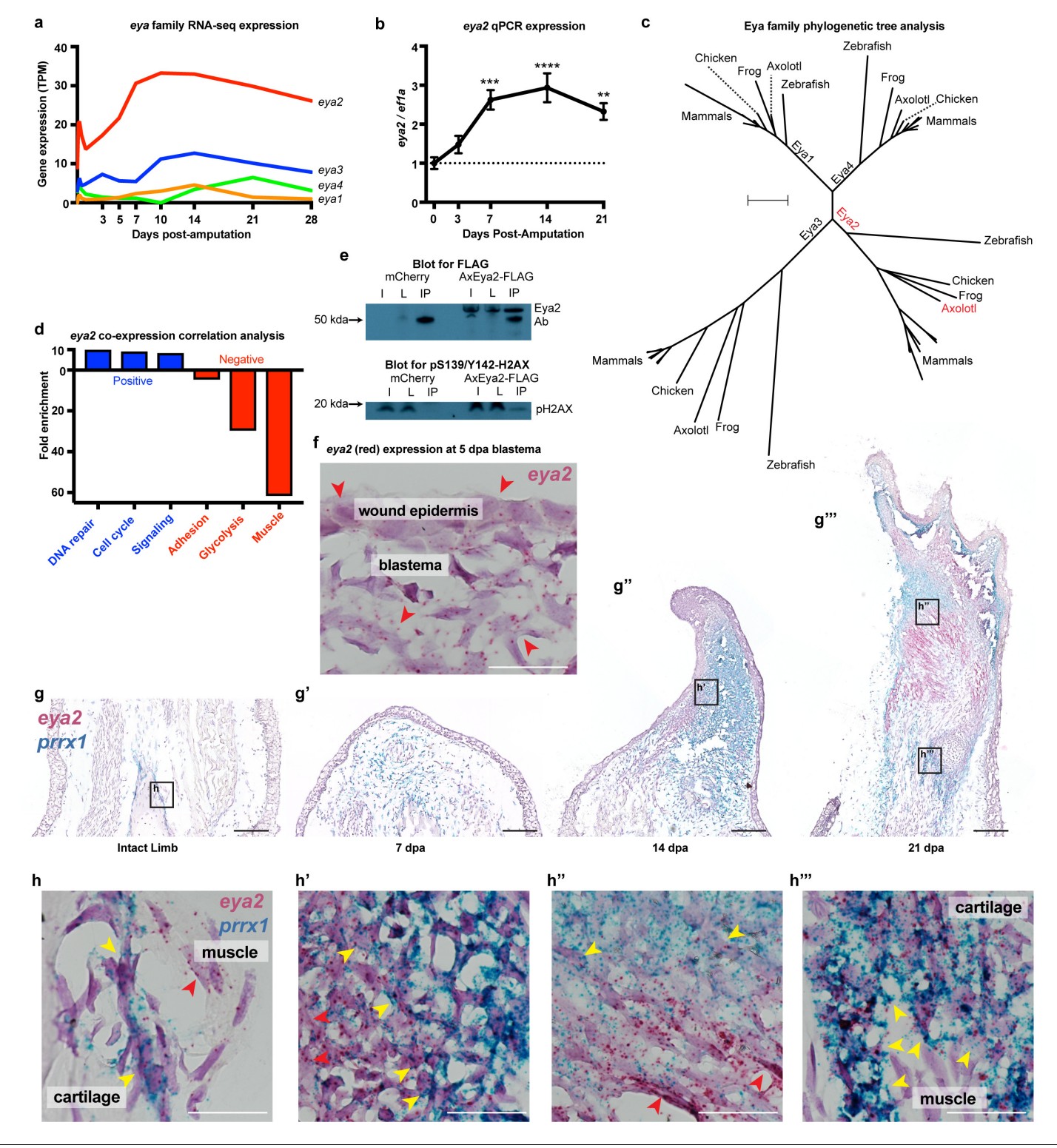

**Figure 2.** *Eya2* is up-regulated during limb regeneration, expressed by limb progenitors and associated with cell cycle and DNA repair. (**a**) RNA-seq derived gene expression of the *eya* gene family during different time points of axolotl limb regeneration. Note that *eya2* is the most expressed gene of the *eya* gene family and appears up-regulated upon amputation. (**b**) Validation of *eya2* gene expression profile using qPCR. We found that normalized *eya2*/*ef1a* levels are significantly up-regulated following limb amputation (One-way ANOVA: p-value<0.0001, with Bonferroni's multiple comparisons test: 0 dpa vs 7 dpa: p-value<0.001, 0 dpa vs 14 dpa: p-value<0.0001, 0 dpa vs 21 dpa: p-value<0.01), indicating that Eya2 may play an important role during limb regeneration. (**c**) Validation of Eya2 homology with other vertebrate Eya2 proteins using phylogenetic tree analysis. (**d**) Gene Ontology

*Figure 2 continued on next page*

Figure 2 continued

Enrichment Analysis (FDR < 0.05) of genes found to be positively ($R^2$ >0.8, slope >0, blue bars) or negatively ($R^2$ >0.8, slope <0, red bars) associated with *eya2* expression. These data suggest that Eya2 is associated with DNA repair and cell cycle during limb regeneration. (e) Co-immunoprecipitation (Co-IP) experiment using the axolotl Eya2 protein fused with a FLAG tag (AxEya2-FLAG) or mCherry control. Immunoblot against the FLAG tag revealed the axolotl Eya2 protein in the samples transfected with the *Eya2*-FLAG plasmid and not in the control, as expected (blot for FLAG). Immunoblot against the phosphorylated form of H2AX revealed a band in the precipitate derived from the AxEya2-FLAG IP sample and not in the mCherry control (blot for pS139/Y142-H2AX). I: Sample before IP, L: Flow-through after IP, IP: Pulled-down proteins. (f) RNA in situ hybridization for *eya2* (red puncta) in 5 dpa blastema section, Scale Bar: 50 µm. *Eya2* appears to be expressed by most cells in the wound epidermis and blastema. (g–g''') Time course RNA in situ hybridization for *eya2* (red puncta) and *prrx1* (blue puncta) during regeneration. Scale Bars: 250 µm. (g) Intact limb. (g'). 7 dpa early blastema. (g'') 14 dpa mid-late blastema. (g''') 21 dpa palette stage. (h–h''') High magnification inserts of (g–g'') images. Scale Bars: 25 µm. (h) Intact limb. *Eya2* is expressed by muscle cells (red arrow) and *prrx1*+ periosteal cells (yellow) which are known cartilage progenitors. (h') 14 dpa blastema. *Eya2* is robustly expressed in the regenerating muscle (red arrow) and blastema cells positive for *prrx1* (yellow arrow) (h'') 21 dpa regenerate. *Eya2* is expressed in the regenerating muscle (red arrows) while expression in the *prrx1*+ population is getting diminished (yellow arrow). (h''') 21 dpa regenerate-proximal area. *Eya2* is expressed in the newly regenerated *prrx1*+ periosteal cells (yellow arrow), while there is no expression in fully developed muscle. ns denotes p-value>0.05, * denotes p-value<0.05, ** denotes p-value<0.01, *** denotes p-value<0.001, **** denotes p-value<0.0001. Vertical bars on plots represent standard error of the mean (SEM).

The online version of this article includes the following figure supplement(s) for figure 2:

**Figure supplement 1.** Gene Ontology Enrichment Analysis of *eya2* co-expression correlation analysis during limb regeneration.
**Figure supplement 2.** *Eya2* lineage trace analysis.

newly-formed areas containing connective tissue progenitor cells were strongly positive for both *eya2* and *prrx1* (*Figure 2g''' and h'''*). These data suggested that the majority of early-stage limb progenitors express *eya2* during limb regeneration. To further strengthen these findings, we performed a lineage tracing analysis by generating knock-in axolotls expressing the Cre recombinase under the endogenous *eya2* promoter in the previously described Cre reporter background (*Figure 2—figure supplement 2a–c*; *Fei et al., 2017*). Using this method, we showed that muscle, skin, connective tissue and cartilage were labeled during limb development and these labels were sustained during regeneration (*Figure 2—figure supplement 2d–e*). Taken together, *eya2* was found up-regulated during regeneration, expressed by most of early-bud blastema cells, and highly correlated with DNA repair genes, all of which strongly suggests that Eya2 activity is part of the DDR initiated upon limb amputation.

## *Eya2* mutant axolotls have reduced rate of regeneration and slower cell cycle progression

To determine whether DDR is important for limb regeneration, we targeted the *eya2* locus with a specific CRISPR/Cas9 ribonucleoprotein complex and generated *eya2* mutant axolotls (*eya2^{-/-}*) (*Figure 3—figure supplement 1a–b*). Based on the genotyping, the Eya2 mutant protein is predicted to be severely truncated and lacking both the phosphatase and transactivation domains (*Figure 3—figure supplement 1a*). Eya2^{-/-} axolotls developed normal limbs with muscle, bone, and nerves (*Figure 3—figure supplement 2a–d*). The *eya2^{-/-}* animals could be easily identified by the presence of an intra-peritoneal edema which was also correlated with malformed kidneys (*Figure 3—figure supplement 2e*, red box). To determine the effects of ablating the function of Eya2 during limb regeneration, we imaged blastemas and quantified their length over time (*Figure 3a*). The data indicated that *eya2^{-/-}* axolotls have reduced blastema length compared to *eya2^{+/-}* and *eya2^{+/+}* at multiple time points (*Figure 3b and c*). We did not observe haploinsufficiency or major differences between *eya2^{+/+}* and *eya2^{+/-}*, so all subsequent comparisons were focused on *eya2^{-/-}* and *eya2^{+/+}* genotypes. At 30 dpa, a pre-selected time point which captures regeneration before the end of the growth process, we collected the regenerated limbs and visualized their skeletons using alcian blue and alizarin red S stains (*Figure 3d*). To determine the size of the regenerate at 30 dpa, we measured their zeugopod length relative to animal size and found that *eya2^{-/-}* animals regenerated smaller limbs, suggesting that the rate of regeneration was impaired (*Figure 3e*, see also *Figure 3—figure supplement 2c–d* for their size before amputation). We then analyzed their skeletal pattern for completion and found that none of the *eya2^{-/-}* animals (n = 0/18 limbs from nine animals) had a fully-patterned skeleton compared to 17% of *eya2^{+/+}* (n = 3/18 limbs from nine animals, 3 × 2 Contingency Table: p-value<0.05, *Figure 3f*). These data indicated that *eya2^{-/-}* axolotls regenerated limbs more

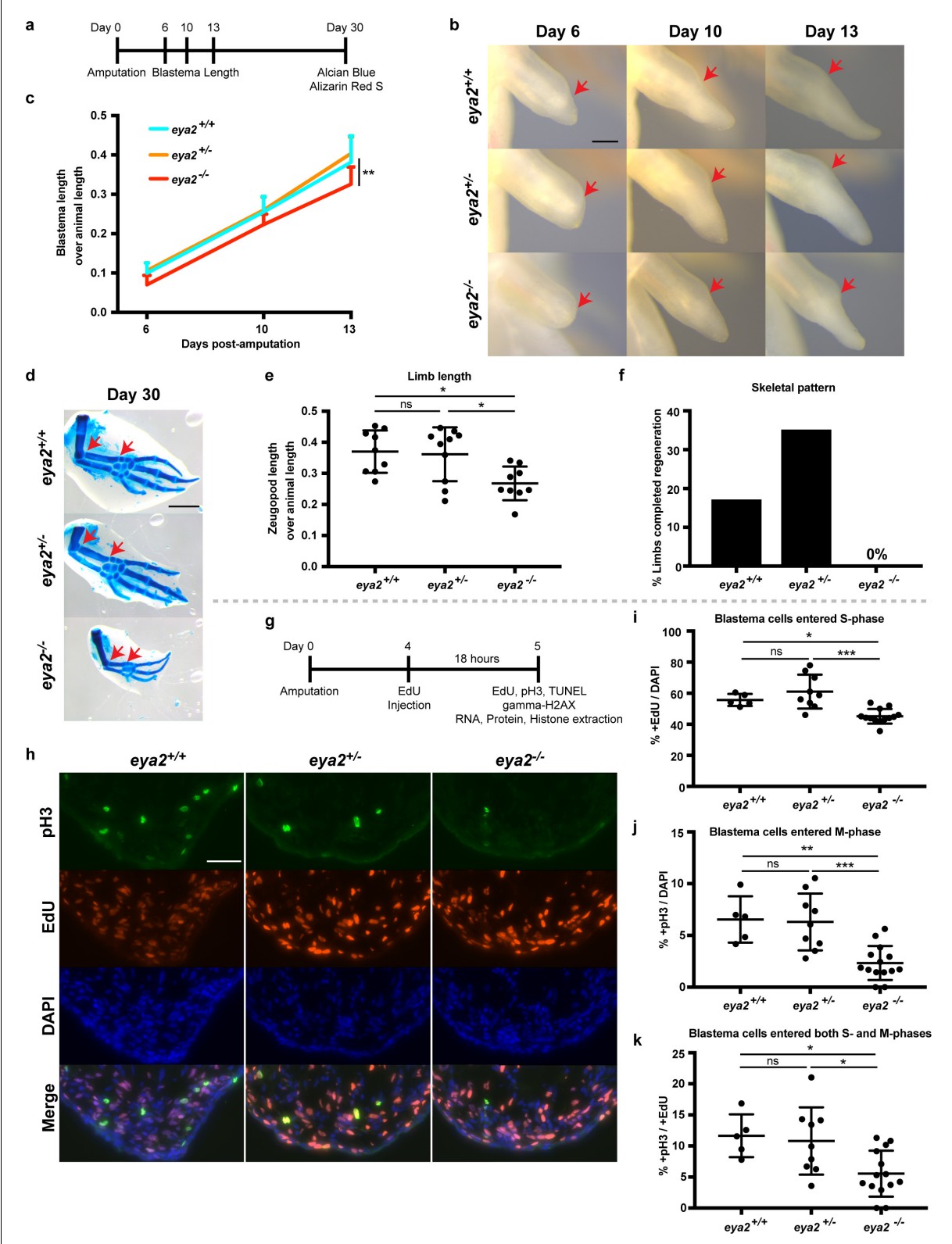

**Figure 3.** CRISPR/Cas9-targeted *eya2* mutant axolotls regenerate slower and have stalled cell cycle. (a) Experimental design to test limb regeneration rates in *eya2⁻/⁻* axolotls. Bright field imaging was conducted at 6, 10, and 13 dpa to determine blastema length overtime. Final limb collection was performed at 30 dpa, time that some control axolotls had all four digits regenerated. 30 dpa limb were processed to visualize their skeleton using alcian blue and alizarin red S. (b) Representative images of blastema at different time points during regeneration. Arrows point at the amputation site.
*Figure 3 continued on next page*

Figure 3 continued

Scale Bar: 0.5 mm. Note that *eya2*[-/-] axolotls have a smaller blastema even at 6 dpa. (**c**) Quantification of blastema length from (**b**). *Eya2*[-/-] axolotls have smaller blastema compared to *eya2*[+/+] and *eya2*[+/-] (Repeated measures two-way ANOVA, interaction: p-value<0.05, time: p-value<0.0001, genotype <0.01; Bonferroni's multiple comparisons tests: 13 dpa *eya2*[+/+] vs 13 dpa *eya2*[-/-]: p-value<0.01 and 13 dpa *eya2*[+/-] vs 13 dpa *eya2*[-/-]: p-value<0.0001). (**d**) Representative images of limb regenerates stained with alcian blue and alizarin red S stains to visualize skeletal elements. 30 dpa regenerates. Red arrows point to the beginning and the end of the zeugopod. Scale Bar: 1 mm. Note the relative smaller limb regenerates of the *eya2*[-/-] axolotls. (**e**), Quantification of zeugopod length from (**d**). *Eya2*[-/-] 30 dpa regenerates are significantly smaller than those from *eya2*[+/+] and *eya2*[+/-] axolotls (One-way ANOVA with Bonferroni's multiple comparisons tests: *eya2*[+/+] vs *eya2*[-/-]: p-value<0.05, *eya2*[+/-] vs *eya2*[-/-]: p-value<0.05). (**f**) Analysis of skeletal pattern from (**d**). Skeletal elements were quantified to determine whether or not the full limb pattern is regenerated. *Eya2*[-/-] animals did not have any limb that had successfully regenerated all skeletal elements (Fisher Exact Probability Test on a $2 \times 3$ table: p-value<0.05). (**g**) Experimental design to test cellular proliferation during regeneration. EdU was injected at 4 dpa and blastema tissues were harvested 18 hr later and processed for immunofluorescence (IF). (**h**) Representative images of blastema sections at 5 dpa stained with pS10-H3, EdU and DAPI. Scale Bar: 100 μm. (**i**) Quantification of percent +EdU cells in the whole blastema. There are less blastema cells entering the cell cycle in *eya2*[-/-] axolotls (Kruskal-Wallis test with Dunn's multiple comparisons test: *eya2*[+/+] vs *eya2*[-/-]: p-value<0.05, *eya2*[+/-] vs *eya2*[-/-]: p-value<0.001). (**j**) Quantification of percent +pH3 cells in the whole blastema. There are less blastema cells actively dividing in *eya2*[-/-] axolotls (One-way ANOVA with Bonferroni's multiple comparisons test: *eya2*[+/+] vs *eya2*[-/-]: p-value<0.01, *eya2*[+/-] vs *eya2*[-/-]: p-value<0.001). (**k**) Quantification of percent +pH3 cells in the +EdU population. There are less blastema cells that entered the cell cycle at the time of EdU pulse and actively dividing at the time of collection in *eya2*[-/-] axolotls (One-way ANOVA with Bonferroni's multiple comparisons test: *eya2*[+/+] vs *eya2*[-/-]: p-value<0.05, *eya2*[+/-] vs *eya2*[-/-]: p-value<0.05). ns denotes p-value>0.05, * denotes p-value<0.05, ** denotes p-value<0.01, *** denotes p-value<0.001, **** denotes p-value<0.0001. Vertical bars on plots represent SEM.

The online version of this article includes the following figure supplement(s) for figure 3:

**Figure supplement 1.** Eya2 mutant animal genotyping.

**Figure supplement 2.** Eya2 mutant animal characterization.

**Figure supplement 3.** Cell cycle regulation in 5 dpa blastemas across *eya2* genotypes and at different time points during limb regeneration.

slowly, impacting the volume of the regenerate as well as the phase of the regenerative process, compared to *eya2*[+/+] siblings.

We then tested whether *eya2*[-/-] animals had an overall slower growth rate and found that even though all genotypes had approximately the same size animals at the time of amputation (*Figure 3—figure supplement 2f*), *eya2*[-/-] animals were slightly longer at 30 dpa (*Figure 3—figure supplement 2g*). These data indicated that the slower growth of the *eya2*[-/-] blastema was due to impairment of the regenerative process itself. We then investigated whether the slower regeneration rate was associated with proliferation deficiencies in limb progenitors. To test this possibility, we performed an 18 hr pulse with the nucleotide analogue EdU and subsequent detection as well as an immunofluorescence assay with the mitotic marker phosphoS10-Histone H3 (pH3) on 5 dpa blastema sections (*Figure 3g and h*). At this time point we had previously observed that *eya2* was widely expressed throughout the blastema (*Figure 2f*). *Eya2*[-/-] blastemas had reduced rates of cell cycle entry indicated by a decrease in EdU[+] cells (*Figure 3i*) and decreased mitotic activity as evidenced by a decrease in pH3[+] blastema cells (*Figure 3j*). To identify whether the rate-limiting factor was S phase entry, which in turn limits the number of cells at the G2 and M phases, or whether there were roadblocks at later phases, we quantified double-positive EdU and pH3 cells. Interestingly, we found a reduced percentage of double EdU+/pH3+ cells in the *eya2*[-/-] blastemas, which indicated that cell cycle progression was also stalled at the G2/M phase (*Figure 3k*).

To further investigate the proliferation deficits observed at the cellular level, we performed western blot analysis of proteins known for their role during the cell cycle (*Figure 3—figure supplement 3a*). We found that the amount of the Cyclin-dependent kinase 2 (CDK2) was reduced in the 5 dpa *eya2*[-/-] blastemas whereas the G2/M-specific Cyclin B1 was increased (CCNB1, *Figure 3—figure supplement 3a*). Given that CDK2 is not degraded at any phase of the cell cycle, its reduced presence indicated that the cell cycle was slower without Eya2 (*Vermeulen et al., 2003*; *Malumbres and Barbacid, 2009*). Interestingly, CDK2 has been shown to stall the cell cycle at the G1 phase when down-regulated which strengthen our EdU data showing G1/S arrest in Eya2 mutant blastemas (*Neganova et al., 2011*). Cyclin B1 protein levels normally increase during the G2 phase and decline following mitosis (*Gong and Ferrell, 2010*; *Gong et al., 2007*). The elevated levels of Cyclin B1 in *eya2*[-/-] blastemas strongly suggest that the cell cycle is stalled at the G2/M DNA damage checkpoint corroborating our pH3/EdU data (*Figure 3k*). Taken together, these data show that ablation of Eya2 transactivation and phosphatase domains has a regeneration phenotype associated with slower cell cycle progression at the two major checkpoint transitions, G1/S and G2/M.

## *Eya2*$^{-/-}$ axolotls have increased levels of γ-H2AX specifically associated with cycling blastema cells

G1/S and G2/M checkpoints are activated upon cellular stress including DNA damage (*Zhou and Elledge, 2000*); thus, we analyzed the phosphorylation status of H2AX as described earlier (*Figure 4a*). We found that 5 dpa blastemas from *eya2*$^{-/-}$ animals had increased levels of relative γ-H2AX compared to those from *eya2*$^{+/+}$ siblings (*Figure 4b*), suggesting that the activated cell cycle checkpoints may be associated with increased levels of genotoxic stress. In contrast, *eya2*$^{+/-}$ blastemas showed increased levels of pY142-H2AX, an outcome that correlated with the reduced Eya2 activity in the heterozygous background (*Figure 4c and d*). Since pY142-H2AX is a direct target of the Eya2 phosphatase domain, the data also indicate that the induced mutations in the *eya2* locus do in fact generate dysfunctional Eya2 mutant protein. However, we did not detect significant changes at pY142-H2AX between *eya2*$^{-/-}$ and *eya2*$^{+/+}$ (*Figure 4c and d*). The discrepancy between *eya2*$^{-/-}$ and *eya2*$^{+/-}$ in the pY142-H2AX levels might be due to compensatory mechanisms that prevents persistently high levels of bi-phosphorylated H2AX, a known cause of cell death (*Cook et al., 2009*). This possibility is supported by TUNEL staining in which we could not detect blastema cells undergoing apoptosis in *eya2*$^{-/-}$ animals (*Figure 4—figure supplement 1a*). We then focused on understanding why *eya2*$^{-/-}$ axolotls have increased γ-H2AX levels. To investigate this, we performed immunohistochemistry against γ-H2AX on tissue sections and identified a robust stain at the skin and wound epidermis (*Figure 4—figure supplement 1a'*). Its high fluorescence intensity throughout the skin did not allow us to detect the characteristic nuclear γ-H2AX foci (*Nakamura et al., 2010*; *Figure 4e and f*). We then focused on blastema cells in which we were able to detect and quantify nuclear γ-H2AX foci per area in EdU+ and EdU- cells (*Figure 4g*). Our analysis revealed that EdU+ cells had more γ-H2AX foci per nuclear area than EdU- cells, irrespective of genotype (*Figure 4h*, Two-way ANOVA, Row factor p-value<0.0001, F(1,30) = 62.39, DF = 1). These data support our initial findings that during limb regeneration total γ-H2AX levels were elevated to alleviate replication stress during blastema cell proliferation (*Figure 1h*, *Figure 1—figure supplement 3*). Notably, our data revealed that EdU+ *eya2*$^{-/-}$ blastema cells had significantly more γ-H2AX foci per area compared to those of *eya2*$^{+/+}$ (*Figure 4h*). This indicated that when blastema cells without Eya2 activity enter S-phase, they acquire more hallmarks of genotoxic stress, which also corroborated our earlier data showing cell cycle arrest at the G2/M checkpoint (*Figure 3k*, *Figure 3—figure supplement 3a*). We further identified that the increased presence of γ-H2AX foci was due to more cells having a high number of foci per area and fewer cells having low number of foci (*Figure 4i*). We also found that EdU+ cells were bigger in size compared to EdU- cells across all genotypes (*Figure 4—figure supplement 1b*, two-way ANOVA, Row Factor p-value=0.004, F(1,30)=16.13, DF = 1); however, differences in the number of γ-H2AX foci per area of the nucleus within the EdU+ population were not due to variations in the cell size (*Figure 4—figure supplement 1b–c*). In addition, when foci from all blastema cells were taken into consideration, there was no difference across genotypes; thus, the effect was found to be specific to cycling blastema cells (*Figure 4—figure supplement 1d*). Since *eya2*$^{-/-}$ blastema cells showed increased levels of the DNA damage marker γ-H2AX they might have also accumulated DNA damage. To test this, we performed comet assays on blastema cells from all three genotypes and found that ablation of Eya2's transactivation and phosphatase domains does not affect the amount of DNA damage; hence, this phenotype appears specific to how DDR is regulated (*Figure 4—figure supplement 1e*). Taken together, these data revealed that functionally ablating one of the components of DDR, Eya2, reduced regeneration rate, which was associated with a genotoxic stress-like phenotype including slower cell cycle dynamics and increased levels of γ-H2AX in proliferating blastema cells.

## RNAseq analysis of *eya2*$^{-/-}$ blastema during limb regeneration validates cell cycle arrest, H2AX de-regulation and elevated DDR

In order to better understand the connection between Eya2, DDR and regeneration, we performed RNAseq in intact and blastema tissues at different timepoints upon amputation from all three *eya2* genotypes. Computational analysis revealed 4461 genes to be differentially regulated (q < 0.05) between genotypes (*Supplementary file 3*). From these, most of them appeared to be down-regulated in *eya2*$^{-/-}$ compared to *eya2*$^{+/+}$ tissues. GO analysis revealed terms previously found to be enriched in the dataset positively-associated with *eya2* expression (*Figure 5—figure supplement 1*

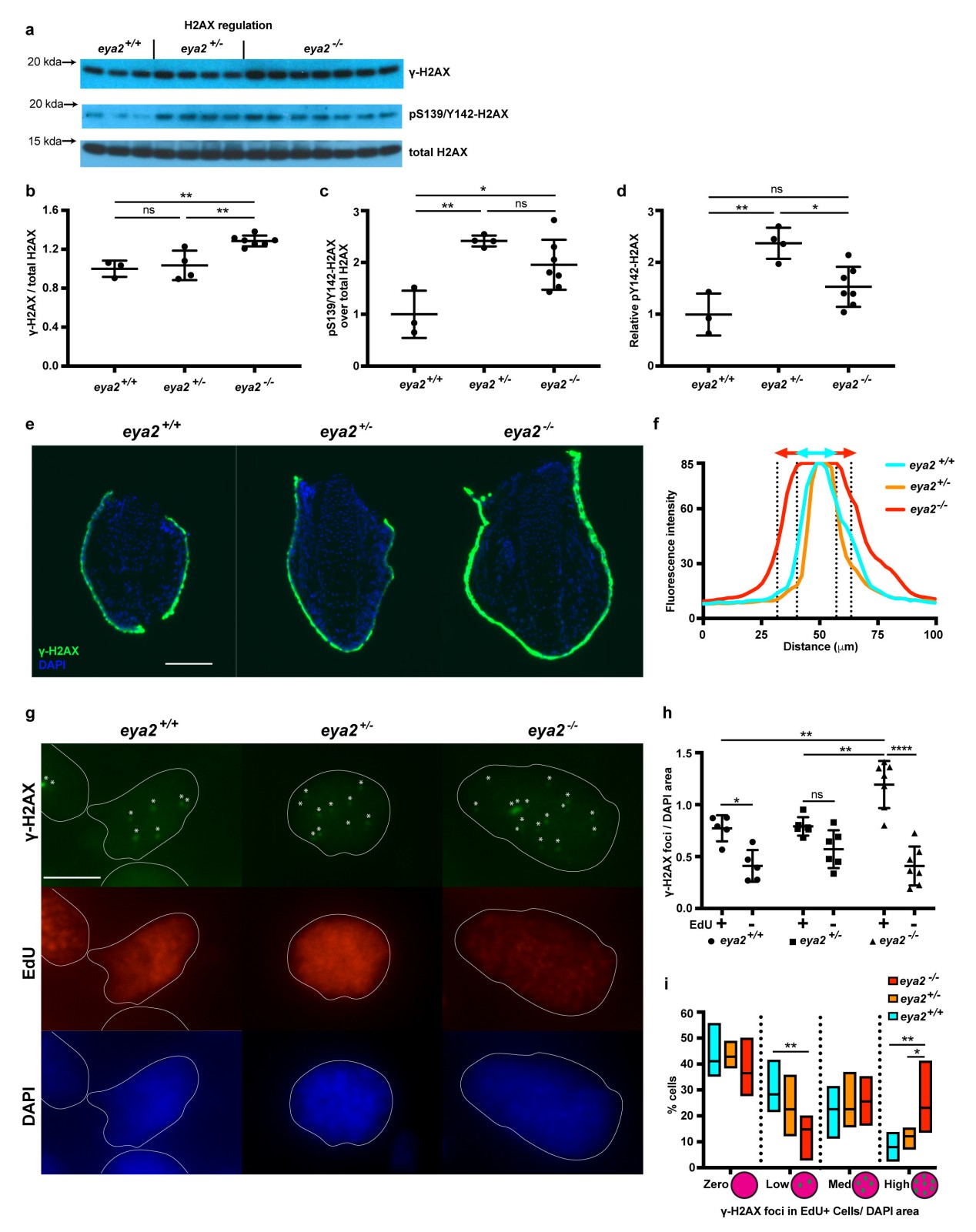

**Figure 4.** Proliferating *eya2⁻ᐟ⁻* blastema cells have increased γ-H2AX levels. (a) Western blot for H2AX and its phosphorylation status in 5 dpa blastema tissue. Note the robust up-regulation of γ-H2AX in *eya2⁻ᐟ⁻* tissues and the up-regulation of pS139/Y142-H2AX signal in *eya2⁺ᐟ⁻* tissues. (b,c,d) Quantification of (a). (b) Relative γ-H2AX levels across genotypes. *Eya2⁻ᐟ⁻* blastema have significantly more relative γ-H2AX levels, indicating higher genotoxic stress signals (One-way ANOVA with Bonferroni's multiple comparisons test: *eya2⁺ᐟ⁺* vs *eya2⁻ᐟ⁻*: p-value<0.01, *eya2⁺ᐟ⁻* vs *eya2⁻ᐟ⁻*:

*Figure 4 continued on next page*

Figure 4 continued

p-value<0.01). (c) Relative pS139/Y142-H2AX levels across genotypes. *Eya2*[+/+] blastema have significant less relative pS139/Y142-H2AX signal than *eya2*[+/-] and *eya2*[-/-] blastema (One-way ANOVA with Bonferroni's multiple comparisons test: *eya2*[+/+] vs *eya2*[+/-]: p-value<0.01, *eya2*[+/+] vs *eya2*[-/-]: p-value<0.05). (d) Relative pY142-H2AX levels across genotypes. *Eya2*[+/-] blastema have significantly more relative pY142-H2AX levels than *eya2*[+/+] and *eya2*[-/-] (One-way ANOVA with Bonferroni's multiple comparisons test: *eya2*[+/-] vs *eya2*[+/+]: p-value<0.01, *eya2*[+/-] vs *eya2*[-/-]: p-value<0.05). (e) Representative images of γ-H2AX immunofluorescence on 5 dpa blastema sections at low magnification. Note the robust γ-H2AX signal intensity in *eya2*[-/-] skin. Scale Bar: 150 μm. (f) Quantification γ-H2AX intensity across the skin in each genotype from (e). *Eya2*[-/-] axolotls have significantly more γ-H2AX in their skin compared to *eya2*[+/-] and *eya2*[+/+] (Two-way ANOVA interaction: p-value<0.0001, genotype: p-value<0.0001). (g) Representative images of blastema sections stained for DAPI (blue), γ-H2AX (green) and EdU (red). Stars (*) point to nuclear γ-H2AX foci, white lines denote the boundaries of the nucleus. Scale Bar: 10 μm. (h) Quantification of γ-H2AX foci per area of nucleus across different genotypes and proliferation status from (g). Blastema cells that have entered the cell cycle have significantly more γ-H2AX foci per area of nucleus irrespective of genotype (Two-way ANOVA: EdU: p-value<0.0001). EdU+ *eya2*[-/-] blastema cells have significantly increased numbers of γ-H2AX foci per area of nucleus compared to those from *eya2*[+/+] and *eya2*[+/-] blastema (Two-way ANOVA interaction: p-value<0.001, with Bonferroni's multiple comparisons test: EdU+ *eya2*[-/-] vs EdU+ *eya2*[+/+]: p-value<0.01, EdU+ *eya2*[-/-] vs EdU+ *eya2*[+/-]: p-value<0.01). (i) Percent of cells in the +EdU blastema population in different categories of γ-H2AX foci concentrations. The data indicate that the same fraction of cells have γ-H2AX foci irrespective of genotype (Two-way ANOVA with Bonferroni's multiple comparisons test: for γ-H2AX zero concentration: *eya2*[+/+] vs *eya2*[+/-]: p-value>0.05, *eya2*[+/+] vs *eya2*[-/-]: p-value>0.05, *eya2*[+/-] vs *eya2*[-/-]: p-value>0.05) but *eya2*[-/-] blastema cells (red bars) have more cells with high and less cells with low γ-H2AX concentration (Two-way ANOVA with Bonferroni's multiple comparisons test: for low γ-H2AX concentration: *eya2*[-/-] vs *eya2*[+/+]: p-value<0.01, for high γ-H2AX concentration: *eya2*[-/-] vs *eya2*[+/+]: p-value<0.01, *eya2*[-/-] vs *eya2*[+/-]: p-value<0.05). ns denotes p-value>0.05, * denotes p-value<0.05, ** denotes p-value<0.01, *** denotes p-value<0.001, **** denotes p-value<0.0001. Vertical bars on plots represent SEM.

The online version of this article includes the following figure supplement(s) for figure 4:

**Figure supplement 1.** TUNEL assay on Eya2 mutant blastema sections, DNA damage quantification in 5 dpa blastema cells and parameters associated with *Figure 4h*.

vs *Figure 2—figure supplement 1*, *Supplementary file 2* vs *Supplementary file 3*). This indicated that perturbating the normal activity of Eya2 disrupted its already predicted functions during regeneration. For instance, GO terms associated with cell cycle, like anaphase-promoting complex-dependent catabolic process, were found significantly different among the genotypes (*Figure 5—figure supplement 1a*). Focusing on the expression of genes associated with cell cycle progression, we found them significantly down-regulated in *eya2*[-/-] tissues, especially at the late-bud blastema (14 dpa, *Figure 5a*). Gene network analysis showed similar results, with genes associated with DNA replication to be enriched in the dataset found down-regulated in *eya2*[-/-] tissues (*Figure 5—figure supplement 1b*, purple nodes). The most over-represented terms were genes associated with metabolic processes and translation (*Figure 5—figure supplement 1b*, yellow and green nodes and *Supplementary file 3*), which previous studies have implicated as a sign of stress (*Paschen et al., 2007*; *Liu and Qian, 2014*; *Lindqvist et al., 2018*). To test whether this was at least in part related to genotoxic stress, we analyzed the expression of genes associated with DDR and found that they were up-regulated in *eya2*[-/-] blastemas versus *eya2*[+/-] and *eya2*[+/+] (*Figure 5b* and *Supplementary file 1*). These data indicate an elevated DNA damage response in *eya2*[-/-] mutant blastema cells, which corroborates our previous molecular and histological data showing decreased cell cycle progression and elevated γ-H2AX levels (*Figure 4h*). On the other hand, the most over-represented terms on the dataset with genes up-regulated in *eya2*[-/-] tissues were regulation of biological quality, a GO term that encapsulates genes related to driving cellular function to homeostasis (*Figure 5—figure supplement 1c*, red nodes and *Supplementary file 3*) which may be part of a mechanism that compensates for the loss of Eya2 to alleviate the elevated DDR and stress. To better understand this process, we data-mined the expression of genes associated with the *eya* gene family, H2AX regulators and DDR. We found the expression of *eya2* itself to be significantly up-regulated in the *eya2*[-/-] tissues, while the rest of the *eya* family remained unchanged, revealing a potential feedback loop specific to *eya2* (*Figure 5c*). Genes known to regulate H2AX post-translational modifications were found down-regulated which indicated a potential negative feedback loop that could ameliorate the elevated γ-H2AX levels and dampen its effects (*Figure 5d*). To test whether the absence of Eya2's transactivation and phosphatase domains might lead to elimination of certain cell population in the blastema, we data-mined the expression of genes previously found to be specifically enriched in different blastema cell populations (*Gerber et al., 2018*; *Leigh et al., 2018*). Genes that define macrophages (APOE), wound epidermis (K1C12) and fibroblasts (DPT, LUM, PRRX1) were not found differentially expressed across genotypes, with the exception of the

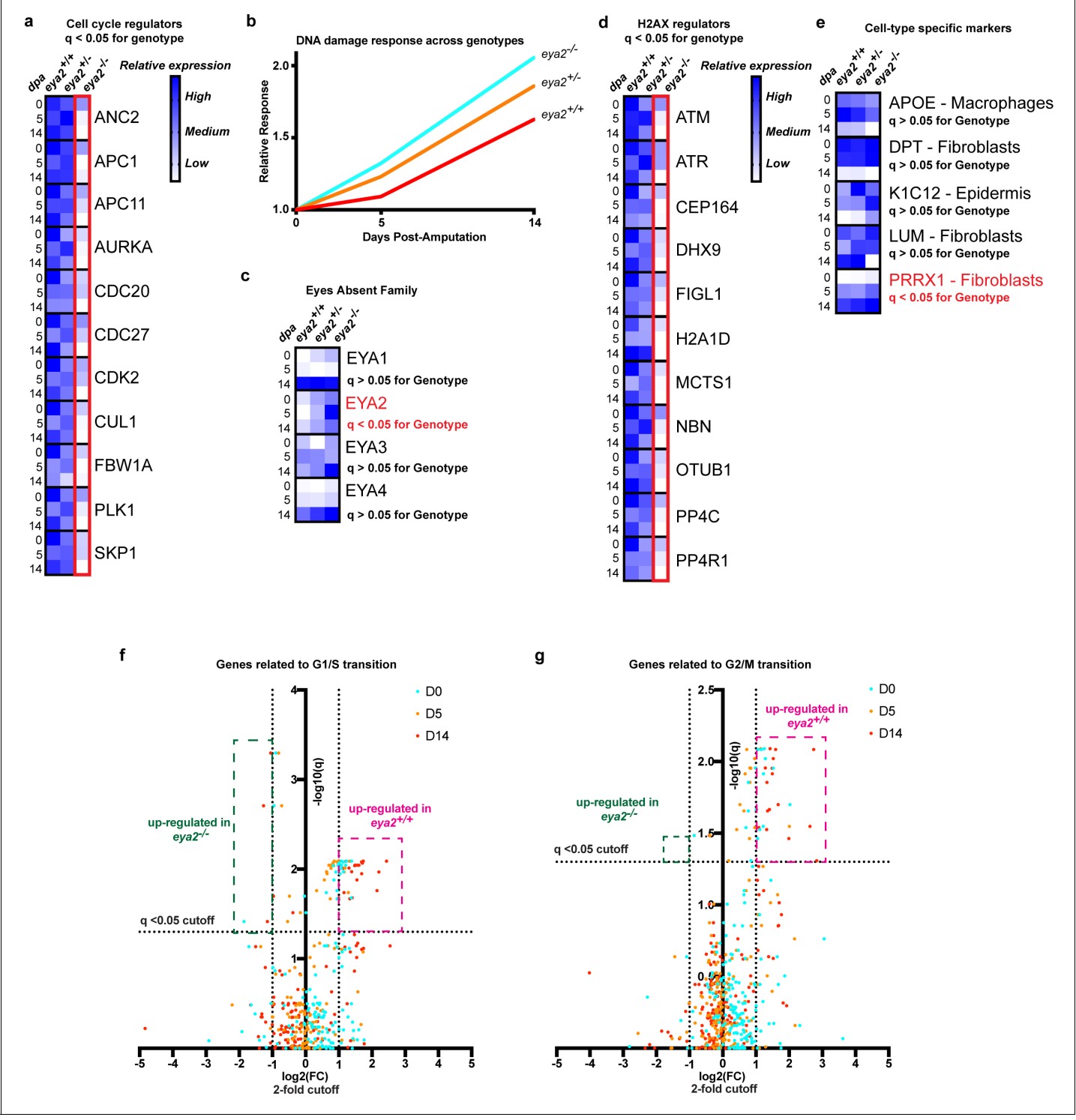

**Figure 5.** RNAseq analysis of *eya2⁻/⁻* tissues confirms the connection between Eya2, DDR and cell cycle progression. (**a**) Relative expression of known cell cycle regulators found differentially regulated (q-value <0.05) across genotypes using RNAseq. *Eya2⁻/⁻* blastema tissues have significantly down-regulated genes associated with cell cycle progression corroborating our previous proliferation data. (**b**) DNA damage response across all genotypes during limb regeneration at 0, 5 and 14 dpa. Note that *eya2⁻/⁻* blastema tissues have significantly elevated the expression of genes associated with DDR indicating a robust genotoxic stress signal. (**c**) Relative expression of the *eya* gene family across genotypes and timepoints. Note that the only differentially regulated gene found using RNAseq analysis (q-value <0.05) was *eya2* itself which appears up-regulated in the *eya2⁻/⁻* background. (**d**) Relative expression of genes known to regulate H2AX post-translational modifications. *Eya2⁻/⁻* blastema tissues have significantly down-regulated these genes (q-value <0.05) indicating a potential molecular mechanism that compensates for the elevated levels of γ-H2AX that persist in absence of Eya2

*Figure 5 continued on next page*

Figure 5 continued

activity. (**e**) Relative expression of genes known to mark specific cell types during axolotl limb regeneration. Expression of macrophage, fibroblast and epidermis markers do not appear to be affected by the absence of Eya2 activity (q-value >0.05). Interestingly, the connective tissue progenitor marker *prrx1* appears up-regulated in the *eya2⁻/⁻* blastemas (q-value <0.05). (**f**) Volcano plot with genes known for their role in G1/S transition of the cell cycle. The pink area depicts genes up-regulated in *eya2⁺/⁺* samples whereas the green area depicts genes up-regulated in *eya2⁻/⁻* samples. Note that most of the genes significantly up-regulated beyond the cutoffs (q < 0.05, fold change (FC) >2) illustrated as dotted lines are in the *eya2⁺/⁺* samples indicating faster cell cycle progression. (**g**) Volcano plot with genes known for their role in G2/M transition of the cell cycle. The pink area depicts genes up-regulated in *eya2⁺/⁺* samples whereas the green area depicts genes up-regulated in *eya2⁻/⁻* samples. Note that most of the genes significantly up-regulated beyond the cutoffs (q < 0.05, fold change (FC) >2) illustrated as dotted lines are in the *eya2⁺/⁺* samples indicating faster cell cycle progression.

The online version of this article includes the following figure supplement(s) for figure 5:

**Figure supplement 1.** GO and network enrichment analysis of biological processes from genes deregulated in Eya2 mutant axolotls.

fibroblast marker *prrx1* that appeared up-regulated in the *eya2⁻/⁻* tissues (*Figure 5e*). These data indicated that there was no loss of any particular cell type in the blastema in absence of Eya2 which further supported a broader role in DDR regulation and cell cycle progression for all limb progenitors. Lastly, we isolated genes known to promote G1/S and G2/M checkpoints and visualized their expression in volcano plots (*Liu et al., 2017*). The data showed that *eya2⁻/⁻* blastemas have down-regulated the majority of these genes, further strengthening our hypothesis that the cell cycle is stalled at the G1/S and G2/M checkpoints in absence of Eya2 activity (*Figure 5f and g*). Taken together, RNAseq analysis revealed down-regulation of genes involved in cell cycle progression specifically involved at the G1/S and G2/M checkpoints as well as H2AX deregulation which corroborated our previous molecular data showing reduced EdU/pH3 staining (*Figure 3k*) while H2AX phosphorylation was aberrantly regulated in *eya2* mutant blastema cells (*Figure 4a*). In addition, the RNAseq data revealed up-regulation of genes consistent with an active compensatory mechanism to alleviate stress and the loss of Eya2 which included the up-regulation of the *eya2* gene itself.

## Pharmacological inhibition of Eya2 upon amputation impairs regeneration and confirms its role in DDR

By genetically ablating Eya2's transactivation and phosphatase domains, we were able to demonstrate its importance during DDR and regeneration. However, our RNAseq analysis revealed a potential compensatory mechanism which was ultimately reflected in the ability of *eya2⁻/⁻* axolotls to successfully regenerate limbs, albeit at a slower pace. To investigate whether this compensation could be employed at the time of amputation in wild-type animals, we used a previously characterized pharmacological inhibitor of Eya2 (*Krueger et al., 2014*; *Krueger et al., 2013*) and asked whether we could phenocopy the DDR defects we observed in our *eya2⁻/⁻* axolotls. We first validated by protein structure analysis that axolotl Eya2 was similar to human Eya2 and displayed the three helices previously found to be important for MLS000544460-mediated inhibition (*Krueger et al., 2014*; *Figure 6—figure supplement 1a and b*). Based on the available pharmacokinetic data (*Krueger et al., 2014*), we administered 25 μM MLS000544460 to wild-type *white* axolotls following amputation and studied its effects on blastema length, proliferation, H2AX regulation, and DNA damage (*Figure 6a*). As expected, Eya2 inhibition significantly reduced blastema length compared to DMSO controls at 8 dpa (*Figure 6b and c*). This effect was accompanied by a decrease in EdU+ cells following an 18 hr pulse (*Figure 6d and e*) and an increase in TUNEL+ apoptotic cells (*Figure 6d and f*) in the treated limbs. We have previously shown that γ-H2AX foci accumulate in proliferating limb progenitor cells following amputation (*Figure 4h and i*) and in combination with the Eya2 inactivation, where pY142 removal is inhibited, could explain an increase of bi-phosphorylated H2AX, a known trigger of apoptosis. To corroborate these data, we used western blot analysis of H2AX where we detected increased levels of pY142- while γ-H2AX remained in the baseline (*Figure 6g and h*, *Figure 6—figure supplement 1c and d*). To determine whether increased levels of DNA damage were also present in MLS000544460-treated blastema cells, we performed alkaline comet assays. We found more DNA damage present in blastema cells that had Eya2 activity blocked (*Figure 6i*), indicating that Eya2 is required for proper DDR in this context. Overall, pharmacological inhibition of Eya2 independently confirmed its role as an important modulatory node for DDR during vertebrate regeneration. In addition, the different phenotypes observed between *eya2⁻/⁻* axolotls

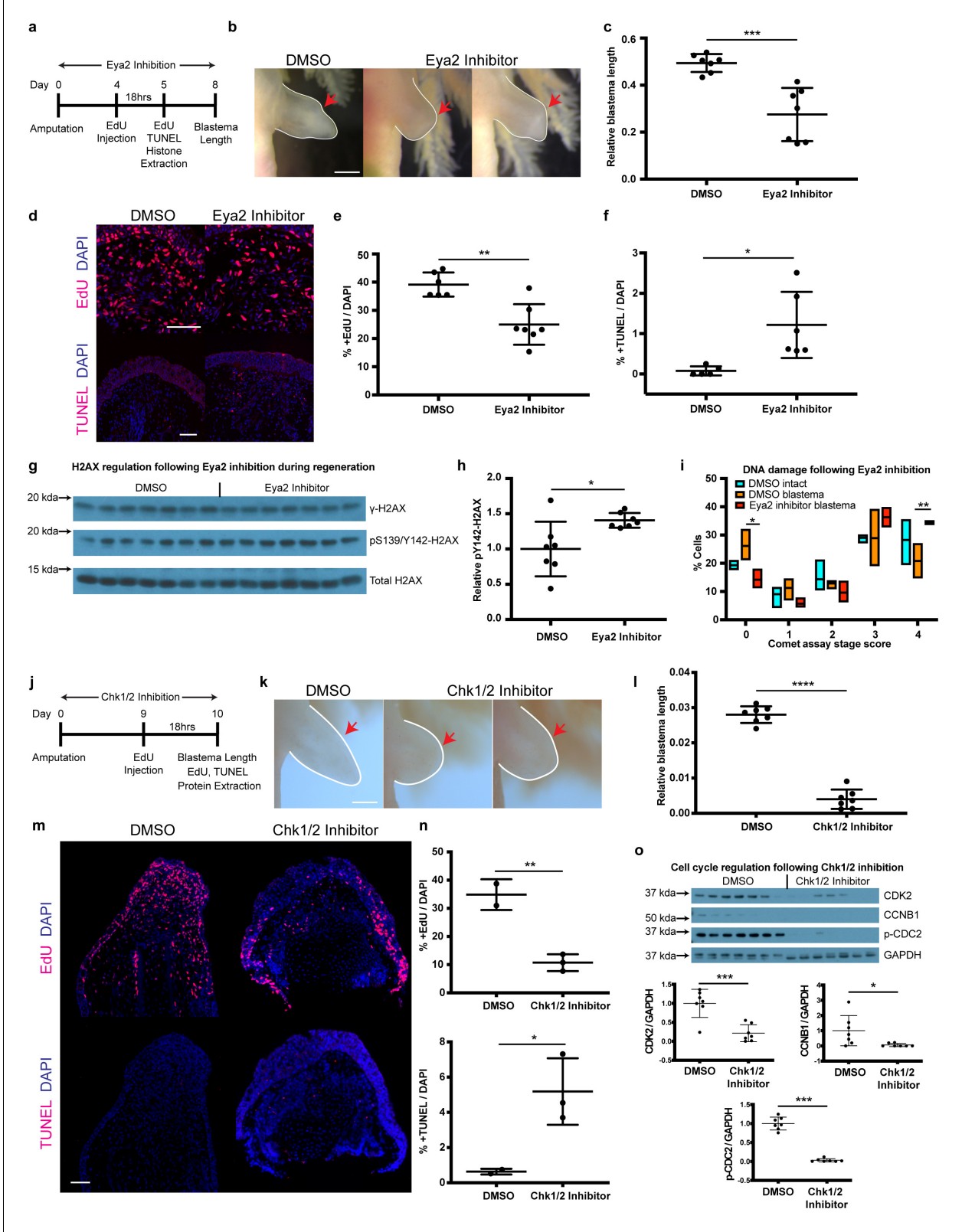

**Figure 6.** Pharmacological inhibition of Eya2 and Chk1/2 impairs regeneration, reduces cell proliferation, increases apoptosis and affects DDR. (a) Experimental design of Eya2 inhibitor treatment: EdU was injected at 4 dpa and tissues were collected for immunofluorescence 18 hr later. EdU, TUNEL, Comet assay and histone extraction was performed at 5 dpa. Blastema length was measured at 8 dpa. (b) Representative images of blastema treated with DMSO and Eya2 inhibitor. Limbs are outlined in white line; red arrows indicate amputation plane. Scale Bar: 2 mm. (c) Quantification of

*Figure 6 continued on next page*

*Figure 6 continued*

blastema length from (**b**). Eya2 Inhibitor-treated blastema tissue is significantly smaller than DMSO-treated ones (Unpaired t test: p-value<0.001). (**d**) Representative images of 5 dpa blastema sections stained for DAPI (blue) and EdU or TUNEL (red) from animals treated with DMSO or Eya2 inhibitor. Scale Bars: 150 µm. (**e**) Quantification of percent +EdU cells from (**d**) top. Eya2 inhibitor-treated blastema tissues have significantly reduced number of cells entering the cell cycle (Unpaired t test: p-value<0.01). (**f**) Quantification of percent +TUNEL cells from (**d**) bottom. Eya2 inhibitor-treated blastema tissues have significantly increased number of cells undergoing apoptosis (Unpaired t test: p-value<0.05). (**g**) Western blot for H2AX and its phosphorylation status in 5 dpa blastema tissues from animals treated with DMSO or Eya2 inhibitor. (**h**) Quantification of relative pY142-H2AX levels from (**g**). Eya2 inhibitor-treated blastema tissues have significantly more relative pY142-H2AX levels indicating that we have successfully inhibited Eya2's phosphatase domain (Unpaired t test: p-value<0.05). (**i**) Percent cells in each comet assay stage score category. A significantly bigger fraction of blastema cells treated with the Eya2 inhibitor show increased levels of DNA damage (Two-way ANOVA with Bonferroni's multiple comparisons test: stage four comet assay score: DMSO Blastema vs Eya2 Inhibitor Blastema: p-value<0.01) while a significantly smaller fraction does not show signs of DNA damage (Two-way ANOVA with Bonferroni's multiple comparisons test: stage 0 comet assay score: DMSO Blastema vs Eya2 Inhibitor Blastema: p-value<0.05). These data indicate that inhibiting Eya2 function upon amputation impairs regeneration rate by reducing cell cycle entry, increasing DNA damage, and triggering apoptosis in blastema cells. (**j**) Experimental design of Chk1/2 inhibitor treatment: EdU was injected at 9 dpa and tissues were collected for immunohistochemistry 18 hr later. At 10 dpa limbs were imaged to quantify blastema length and samples were collected for protein extraction. (**k**) Representative images of blastema treated with DMSO or Chk1/2 inhibitor. Scale Bar: 2 mm. (**l**) Quantification of blastema length from (**k**). Chk1/2 Inhibitor-treated blastema tissue is significantly smaller than DMSO-treated ones (Unpaired t test: p-value<0.0001). (**m**) Representative images of 10 dpa blastema sections stained for DAPI (blue) and EdU or TUNEL (red) from animals treated with DMSO or Chk1/2 inhibitor. Scale Bars: 125 µm. (**n**) Top: Quantification of percent +EdU cells from (**m**) top. Chk1/2 inhibitor-treated blastema tissues have significantly reduced number of cells entering the cell cycle (Unpaired t test: p-value<0.01). Bottom: Quantification of percent +TUNEL cells from (**m**) bottom. Chk1/2 inhibitor-treated blastema tissues have significantly increased number of cells undergoing apoptosis (Unpaired t test: p-value<0.05). (**o**) Western blot analysis of cell cycle regulators following Chk1/2 inhibition. CDK2, CCNB1, p-CDC2 and GAPDH levels were examined. CDK2 levels were significantly reduced in 10 dpa blastema upon Chk1/2 inhibition compared to DMSO controls (Unpaired t test: p-value<0.001). CCNB1 levels were significantly reduced upon Chk1/2 inhibition (Unpaired t test: p-value<0.05). P-CDC2 levels were significantly reduced upon Chk1/2 inhibition (Unpaired t test: p-value<0.05). These data indicate proper regulation of DDR through the major DNA damage checkpoints controlled by Chk1/2 is important for tissue regeneration. ns denotes p-value>0.05, * denotes p-value<0.05, ** denotes p-value<0.01, *** denotes p-value<0.001, **** denotes p-value<0.0001. Vertical bars on plots represent SEM.

The online version of this article includes the following figure supplement(s) for figure 6:

**Figure supplement 1.** Eya2 protein structure visualization and H2AX quantification of *Figure 6g*.

and Eya2 inhibitor-treated axolotls during regeneration, especially the increase in apoptotic cells in the latter, revealed that there is functional compensation that could prevent apoptosis in cells subject to genotoxic stress. This phenomenon has also been observed in other CRISPR/Cas9-mediated models including axolotls (*Junker, 2019*; *Rossi et al., 2015*; *Tsai et al., 2020*).

## Pharmacological inhibition of Chk1 and Chk2 upon amputation impairs regeneration

Our genetic and pharmacological experiments targeting Eya2 have shown that DDR is an important aspect of the regenerative process. To broaden our experiments and the implications of DDR in tissue regeneration, we pharmacologically inhibited the serine/threonine-protein kinases Chk1 and Chk2, which are known to regulate the G1/S and G2/M cell cycle transitions upon DNA damage (*Bartek and Lukas, 2003*; *Smith et al., 2010*). We used the same experimental design as before and quantified blastema length at 10 dpa, EdU+ and TUNEL+ cells in blastema sections, and cell cycle regulators in protein extracts (*Figure 6j*). Our data phenocopy the Eya2 inhibition, showing smaller blastema length (*Figure 6k and l*), reduced levels of EdU+ cells and increased TUNEL+ cells (*Figure 6m and n*) indicating reduced cell cycle progression and increased apoptosis. To validate that the inhibitor deregulated the G1/S and G2/M checkpoints as intended we blotted for CDK2, CCNB1 and p-CDC2, and found them reduced upon Chk1 and Chk2 inhibition (*Figure 6o*). Taken together, by inhibiting another key aspect of DDR, we were able to impede tissue regrowth, further strengthening the hypothesis that proper regulation of genotoxic stress is required for regeneration.

## Discussion

### DNA damage response is an integral process of tissue regeneration

Following limb amputation, a DDR is elevated (*Figure 1a and b*, *Figure 1—figure supplement 1a*, and *Figure 1—figure supplement 2*) which is consistent with previous studies that showed up-

regulation of DNA repair genes during different regenerative processes (*Sousounis et al., 2014*; *Darnet et al., 2019*). However, whether DDR is necessary for successful regeneration or instead is a transcriptional artifact due to the functional annotation of certain genes that have additional roles remained unknown. The differential regulation of H2AX at the RNA, protein and protein modification levels following limb amputation provided evidence that DDR is active during regeneration (*Figure 1b and e–k* and *Figure 1—figure supplement 3*). In fact, at the cellular level, progenitors that have entered the cell cycle displayed increased numbers of γ-H2AX foci in their nucleus, indicative of genotoxic stress, but at the tissue level the relative amounts of γ-H2AX were reduced due to increased levels of unphosphorylated H2AX (*Figure 1i*, *Figure 1—figure supplement 3* and *Figure 4h*). This illustrates the balanced role of DDR during regeneration: efficient DNA repair to ensure integrity but allowing cell cycle progression. Stronger evidence that DDR is an integrated part of the regenerative process was uncovered when we genetically ablated both of the two expected functional domains of one of the DDR's components, Eya2. Our molecular, biochemical and transcriptomic data indicate de-regulation of the DDR process with increased levels of γ-H2AX and further up-regulation of other DDR genes (*Figure 4a*, *Figure 4h* and *Figure 5b*). These resulted in significantly slower cell cycle dynamics with evidence of stalling in the two major DNA damage checkpoints G1/S and G2/M (*Figure 5f*, *Figure 5g*, *Figure 3k*, *Figure 3—figure supplement 3a*) which ultimately impacted the regeneration pace (*Figure 3c*, *Figure 3e* and *Figure 7*).

## Eya2 is up-regulated in proliferating cells to properly control DDR

We found *eya2* to be up-regulated in tissues that undergo regeneration (*Bryant et al., 2017b*; *Sousounis et al., 2014b*) and be preferentially expressed in stem/progenitor cells in intact and regenerating limbs (*Figure 2a*, *Figure 2b Figure 2f*, *Figure 2h* and *Figure 2—figure supplement 2*). Interestingly, Eya2 is also considered a biomarker for certain cancers, indicating that many more types of proliferating cells may up-regulate *eya2* (*Fu et al., 2014*; *Yuan et al., 2016*; *Wen et al., 2017*). Based on its most documented function during DDR, Eya2 prevents apoptosis by dephosphorylating pY142-H2AX which subsequently promotes DNA repair (*Cook et al., 2009*). This mechanism could be active during regeneration and cancer cell proliferation to prevent apoptosis. Our molecular and biochemical data, however, strongly suggest an additional role for Eya2: dampening the DDR in cells following DNA repair. This is especially evident since progenitor cells lacking functional Eya2 have persistent γ-H2AX levels in combination with slower cell cycle dynamics without changes in DNA damage (*Figure 4h*, *Figure 4—figure supplement 1e*; *Tu et al., 2013*). Current literature has not found a role in Eya2 to properly control the return of DDR to baseline and have therefore also not determined whether this is due to its association with H2AX. However, previous studies have identified additional tyrosine residues on H2AX that Eya2 could interact with, as well as crosstalk between them (*Brown et al., 2012*; *Krishnan et al., 2009*; *Liu et al., 2016*). This indicates that the role of Eya2 in resolving the DDR might be based on Y142-H2AX dephosphorylation, which in turn affects S139 phosphorylation in an as-of-now uncharacterized manner or by targeting other tyrosine or threonine resides on H2AX or on other phospho-proteins associated with it. While elucidating this possible mechanism is outside the scope of this study and challenging with the paucity of axolotl antibodies, ongoing studies in mammalian stem cells may uncover it. It is interesting to speculate that cancer cells might utilize Eya2 to bypass proper genotoxic stress signaling, preventing DDR initiation and thereby promoting rapid cell cycle entry, in addition to its already documented role in cell invasion (*Wen et al., 2017*). As Eya proteins have been shown to act as transcriptional co-activators in other systems, considering a possible co-activation role for Eya2 in axolotl limb regeneration will also be important and the subject of future study (*Jemc and Rebay, 2007*). It is also possible that underdeveloped kidneys observed in the *eya2* mutants could lead to a physiological difference between these animals and wild type and that these differences could indirectly impact limb regeneration.

## Genetic and functional compensation for the loss of Eya2 function

Axolotls with one *eya2* mutant allele showed increased amounts of pY142-H2AX levels (*Figure 4d*). This data indicate that the mutant Eya2 protein lacked its normal phosphatase function as predicted by our genotyping of the targeted *eya2* locus. In *eya2*$^{-/-}$ axolotls, since both *eya2* alleles are mutated with the same indels as those from *eya2*$^{+/-}$, we were expecting a further increase in pY142-H2AX

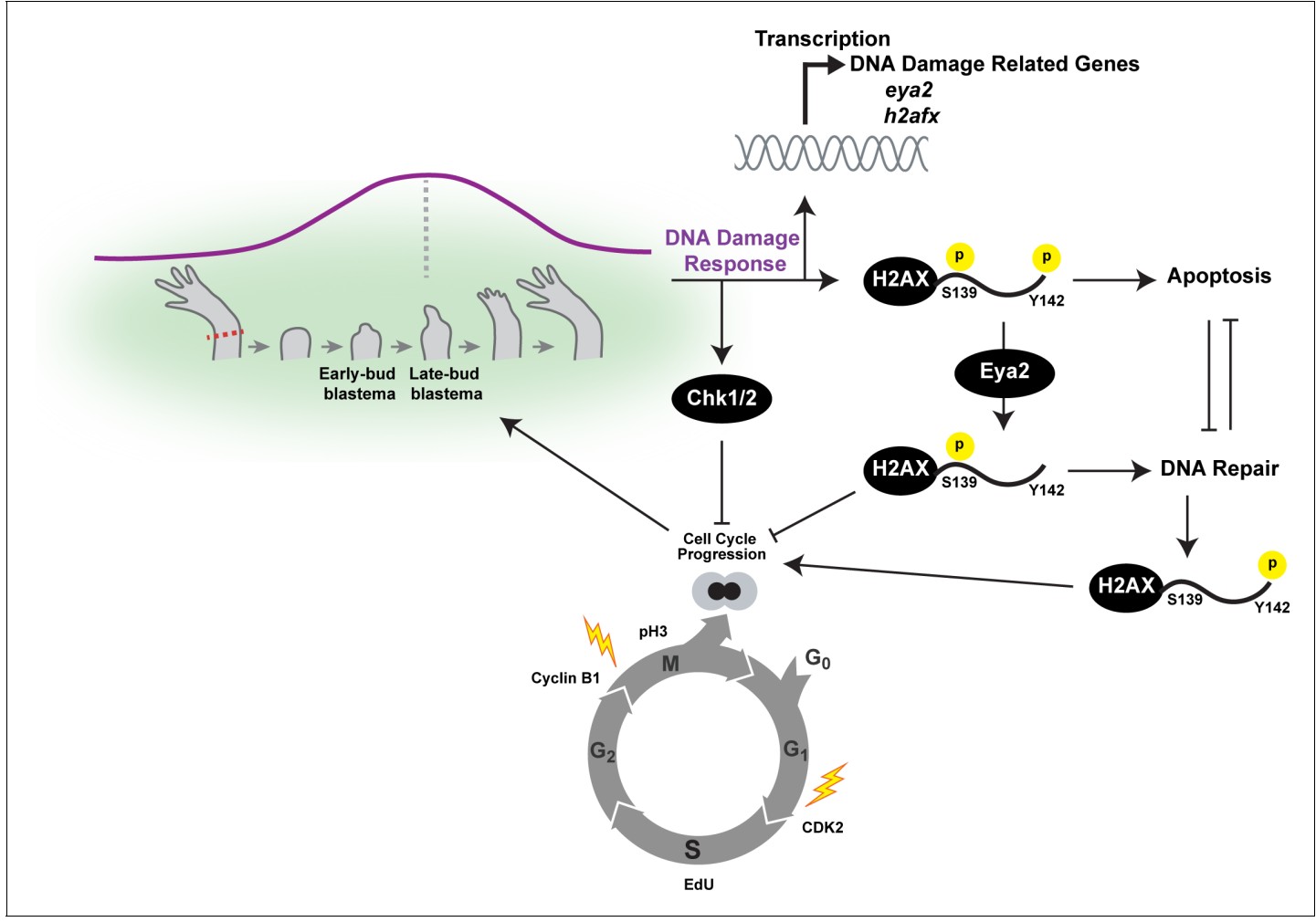

**Figure 7.** Overview of the connections between Eya2, DDR and regeneration. Upon limb amputation, we found that there is up-regulation of genes associated with DNA damage response which peak at the late-bud blastema (purple line). When we inhibited the function of the serine/threonine-protein kinase Chk1 and Chk2 which regulate G1/S and G2/M transitions upon DNA damage, regeneration was inhibited. Interestingly, we found that cell cycle progression was severely impaired with reduction of EdU+ cells and diminishing of cell cycle regulators like CDK2 and Cyclin B1. These data indicated that proper DNA damage response is important for tissue regeneration. This hypothesis was further supported by the dynamic regulation of the DNA damage mediator protein H2AX, especially in progenitor cells that have entered the cell cycle. To further study the role of DDR during regeneration, we genetically ablated one of its components. We chose *eya2* as our target because (1) was found up-regulated during regeneration, and (2) is known to regulate DNA repair by dephosphorylating H2AX at the Y142 amino acid. We found that in the absence of Eya2, there is a robust up-regulation of γ-H2AX levels at the tissue level and the number of nuclear γ-H2AX foci at the cellular level. This indicated an increase in genotoxic stress which was also corroborated transcriptome-wide using RNAseq. Interestingly, the increase in γ-H2AX levels was not associated with an increase of DNA damage accumulating in blastema cells indicating that the observed phenotype was due to deregulation of the DNA damage response by allowing persistent γ-H2AX. We found that this molecular signature resulted in inhibition of the cell cycle progression which ultimately affected the regeneration rate. We further explored the role of Eya2 in regulating DDR and cell cycle progression by utilizing an Eya2 inhibitor. Our data indicate that Eya2 inhibition significantly impaired limb regeneration by de-regulation of H2AX phosphorylation status, decrease in cell cycle progression, higher levels of DNA damage and triggering of apoptosis in blastema cells. Together, our data indicate that Eya2 is a critical modulatory protein that mediates cell cycle progression by properly regulating DDR and its function is important for understanding how highly regenerative organisms like axolotls are able to efficiently regrow tissues without compromising their genomic integrity and cellular potency.

levels and potentially apoptosis (*Smith et al., 2010*). This, however, is not the case since *eya2*$^{-/-}$ tissues have reduced levels of pY142-H2AX than the *eya2*$^{+/-}$ siblings indicating a compensatory mechanism that allowed cells without a functional Eya2 to not undergo apoptosis and axolotls to eventually regenerate normal limbs at a slower pace. We speculate that this compensatory mechanism is acquired during early phases of animal development since without it *eya2*$^{-/-}$ axolotls might be embryonic lethal as many dividing cells would trigger apoptosis. Other studies have also shown

genetic compensation when genes are targeted with CRISPR/Cas9 in a mechanism that is based on mRNA decay (*El-Brolosy et al., 2019*). During this process, there is up-regulation of related genes based on their sequence similarity to the targeted gene. Interestingly, in our CRISPR/Cas9-targeted *eya2* mutants, compensation does not occur with up-regulation of the *eya2* paralogue genes *eya1*, *eya3* and *eya4*. In fact, only *eya2* expression was found significantly up-regulated, strongly suggesting that its transcription is controlled by a feedback loop that senses lower than normal Eya2 activity (*Figure 5c*). Another possibility is that the sequence of *eya2* is not similar enough to its paralogues for the mRNA decay-based mechanism to recognize them. This might also further indicate that the functions of the different *eya* paralogue genes are compartmentalized and likely do not share similar cellular roles in regenerating axolotl limbs. This finding is also supported by the fact that only *eya1* is enriched in presumptive muscle progenitor cells (*Leigh et al., 2018*). Our RNAseq analysis showed that cells might compensate for the loss of Eya2 activity at the functional level by altering the expression of genes associated with the cellular role of Eya2 (*Figure 5a–e*). For example, *eya2* mutant axolotls up-regulate DDR genes during regeneration, which could restore the resolution of genotoxic stress signals (*Figure 5b*). Thus, our model is explained by the combination of a genetic compensation through mRNA degradation and feedback loops which up-regulated *eya2* (*Figure 5c*) and mRNA surveillance mechanisms (*Figure 6—figure supplement 1c*) as well as functional compensation by up-regulation of DDR genes (*Figure 5b*) to facilitate the loss of Eya2 activity. It is interesting to speculate that since Eya2 has many roles during development including regulating DDR to allow cell cycle progression, cells that hard-wire or genetically imprint the mechanism by which functional compensation of Eya2 is achieved would survive better and increase their presence during embryogenesis. This compensatory mechanism is better revealed when we applied the Eya2 inhibitor following amputation. The acute disruption of the DDR, at a time where compensation cannot be readily achieved, blocked regeneration, and triggered apoptosis and DNA damage accumulation (*Figure 6*). This is indicative of the known role of Eya2, to remove Y142 phosphorylation and mediate DNA repair versus apoptosis (*Cook et al., 2009*). Genetic and biological compensation due to CRISPR/Cas9-targeting is a recently-emerging concept seen in axolotls and other animal models (*Junker, 2019*; *Rossi et al., 2015*; *Tsai et al., 2020*). Overall, this compensatory mechanism may have a pivotal role during embryogenesis, regeneration, and evolution. The ability to mask genotypic problems that result in phenotypically normal organisms bears the potential of favorable molecular and genetic mechanisms to build resiliency under stress conditions like those experienced upon injury and regeneration.

## Materials and methods

### Animal handling and procedures

All experiments involving animals were performed according to IACUC protocol #2016N000369 at Brigham and Women's Hospital and IACUC protocol #19-02-346 at Harvard University. Throughout the study, white *Ambystoma mexicanum* axolotls were used and maintained at 68°F as described in *Bryant et al. (2017b)*. All experimental procedures were conducted while animals were anesthetized in 0.1% tricaine. All limb amputations were conducted at the mid-stylopod level. Tail amputations were conducted approximately 1 cm from the tip. Blastemas to be used in western blot, qPCR, RNA-seq or comet assay were removed from the previous amputation plane. Blastemas to be used for immunohistochemistry and in situ hybridization were removed slightly proximal to the previous amputation plane to include visible stump tissues in sections for better identification of the blastema. All animals were left to recover overnight in 0.5% sulfamerazine. *Eya2-/-* axolotls accumulate visible intra-peritoneal fluids (edema) starting approximately 30–40 days post-hatching. To minimize confounding effects due to the edema, all experiments were performed following the complete development of forelimbs at approximately 45 days post-hatching. For the study of limb regeneration rate, animals were anesthetized and placed under a Leica M165 FC dissecting scope equipped with Leica DFC310 FX camera for bright field imaging of blastema growth. Blastema length was measured using ImageJ. For the study of limb regeneration following Eya2 inhibition, animals were treated with 25 μM MLS000544460 (from a 25 mM stock diluted with DMSO, #531085, Calbiochem) or DMSO (1:1000). For the study of limb regeneration following Chk1/2 inhibition, animals were treated with 10 μM AZD7762 (from a 10 mM stock diluted with DMSO, #S1532, Selleckchem) or

DMSO (1:1000). Drug treatments were replenished daily by full water change for the duration of the experiment. When applicable, animals were injected with 20 μl 400 μM EdU per gram of animal weight 18 hr prior to limb collection.

## Skeletal preparations

To analyze the skeletal pattern and the size of intact or 30 dpa regenerated limbs, animals were anesthetized and whole limbs were removed using scissors. Limb samples were incubated in 95% ethanol overnight at room temperature followed by 100% acetone at the same conditions. The skeleton was visualized with alcian blue and alizarin red S stains followed by de-stain of non-skeletal tissue with potassium hydroxide. Images were captured using a Leica M165 FC equipped with Leica DFC310 FX camera. Quantification of the zeugopod length and skeletal pattern was conducted using ImageJ according to *Bryant et al. (2017b)*. Fully patterned limbs were considered those that had 4 digits, 9 phalanges/metacarpals and 8 carpals.

## Comet assay

5Single-cell gel electrophoresis (comet assay) was conducted according to manufacturer's recommendations with minor modifications (Cellbiolabs). Briefly, cells were mechanically dissociated from the tissue with scissors in ice-cold 0.7x PBS/20 mM EDTA. Experiments were conducted in the dark or dim light at 4°C to minimize the introduction of DNA damage. Supernatant containing single cells was isolated and centrifuged at 700xg for 2 min. Pellet was re-suspended at 200,000 cells per ml with ice-cold 0.7x PBS. Cells were then mixed with melted OxiSelect comet agarose in 1:10 ratio and let to solidify on the OxiSelect comet slides for 15 min. Lysis was performed for 1 hr or overnight. Alkaline unwinding was performed for 30 min. Alkaline electrophoresis was performed for 30 min at 1 Volt per cm of electrode distance adjusted to 300mAmps. Slides were then rinsed thrice with ice-cold deionized water for 2 min followed by 5 min incubation with 70% Ethanol and left to air-dry at room temperature. DNA was stained with Vista Green DNA dye for 15 min at room temperature. Comets were visualized using an EVOS FL Auto 2 Cell imaging system. The comet stage of approximately 100 cells per sample was determined as previously described (*Azqueta and Shaposhnikov, 2011*) and based on *Figure 1—figure supplement 1b*. At least three independent samples were used per treatment or genotype. 14 dpa blastemas were treated with UV by placing them in 0.7x PBS under a G30T8 UV lamp for 5 min (*Figure 1c*). Isolated cells for the other comet assays were from intact tissue (0 dpa) or 5 dpa blastema, from wild-type animals or *eya2* mutants, or following treatments with MLS000544460 or DMSO.

## Protein and histone extraction and analysis using western blot

Histone extraction was performed according to manufacturer recommendations (ab113476, Abcam). Protein extraction was performed using RIPA buffer supplemented with protease and phosphatase inhibitors. Histone/protein electrophoresis was performed using NuPage 4–15% Bis Tris Gel with MES-SDS running buffer (Boston BioProducts). 1000 ng total protein measured by bicinchoninic acid assay (BCA) were used for western blots using samples following histone extraction; the loading of equal total histones is used to infer total amounts of H2AX and γ-H2AX in *Figure 1*. 20000 ng total protein was used for western blots in *Figure 3—figure supplement 3b*. 11897.68 ng total protein was used for western blots in *Figure 3—figure supplement 3a*. 5500 ng total protein was used for western blots in *Figure 6o*. Differences in the amount of protein used for western blots were based on sample availability and normalization across samples based on the BCA values. Protein transfer was performed with 20% methanol Tris-Glycine on a 0.2 μm nitrocellulose membrane (Bio Rad). Blots were blocked with 5% BSA 0.1% Tween 1X TBS (TBST) or 5% milk powder in TBST for 1 hr at room temperature and then blotted overnight with primary antibody on blocking buffer at 4°C. Blots were washed thrice with TBST and incubated with HRP-conjugated secondary antibody for 1 hr at room temperature. Blots were then washed thrice with TBST and incubated with Enhanced chemiluminescence (ECL) solution for 1 min before being exposed to the X-ray film. Antibodies used and dilutions were: phospho-histone H2A.X (Ser139/Tyr142), 1:1000 (#5438, Cell Signaling Technology); anti-gamma H2A.X (phospho S139), 1:2000 (#ab11174, Abcam); anti-Histone H2A.X, 1:5000 (#ab180552, Abcam); anti-Cdk2 (78B2) 1:1000 (#2546S, Cell Signaling Technology), phospho cdc2 (Tyr15) (10A11) 1:1000 (#4539S, Cell Signaling Technology), Cyclin B1 (D5C10) 1:1000 (#12231S, Cell

Signaling Technology), GAPDH 1:20000 (#G8795, Sigma), goat anti-mouse HRP, 1:10000 (#103-035-155, Jackson ImmunoResearch); goat anti-rabbit HRP, 1:5000 (#1721019, Bio-Rad). Films were scanned at 600dpi, image exposure was inverted and converted to grey-scale in Adobe Photoshop before protein was quantified using ImageJ. To calculate the relative amounts of pY142-H2AX, we subtracted the relative values of γ-H2AX from that of S139/Y142-H2AX. The results represent the relative amounts of phosphorylated Tyr142 depicting either mono-pY142-H2AX or bi-pS139/pY142-H2AX.

## Immunoprecipitation (IP)

Human 293 T cells were cultured in 10% DMEM at 37°C and transfected with pCAG-mCherry or pCAG-AxEya2-FLAG using Lipofectamine 3000 (Thermo Fisher) following the manufacturer's recommendations. Cells were lysed using RIPA buffer (Thermo Fisher) and IP was performed using an anti-FLAG antibody (#F1804, Sigma) and Dynabeads Protein G (Thermo Fisher) according to manufacturer's recommendations. Western blot was performed as described before. Antibodies used were: anti-FLAG 1:2000 (#F1804, Sigma) and anti-phospho-histone H2A.X (Ser139/Tyr142) 1:1000 (#5438, Cell Signaling Technology). As controls to the IP, the same volumes of the initial protein lysate and of the flow-through were used alongside the proteins pulled-down by the antibody.

## Tissue preparation for paraffin sectioning and H & E staining

Whole animals from different *eya2* genotypes were euthanized and fixed in 4% paraformaldehyde in PBS overnight at 4°C. Tissues were dehydrated in an ethanol series before placed in xylene and embedded in paraffin using a Leica EG 1160. Cross-sections were acquired using a Thermo Scientific MicroM HM 355S microtome. Approximately two sections were placed in slides from every 10 acquired throughout the body axis in order to obtain a representative collection of all internal organs and tissues. Paraffin sections were washed with xylene and ethanol before hematoxylin and eosin staining was performed. Tiled images were obtained and stitched using an Evos FL Auto two imaging system.

## Tissue preparation for immunofluorescence and in situ hybridization

Intact or blastema tissue was extracted using scissors and fixed in 4% paraformaldehyde in DEPC-treated PBS overnight at 4°C. Tissue was then cryopreserved in a sucrose gradient and then incubated in 30% sucrose/PBS at 4°C overnight. Tissue was then embedded in Tissue-Tek O.C.T. Compound (Sakura) and was stored at −80°C. Sectioning was performed using a Leica CM 1950 cryotome at 16 μm thickness. Sections were stored at −80°C until use. Blastema tissues derived from *eya2^-/-*, *eya2^+/-* and *eya2^+/+* animals were embedded in the same block to ensure identical preparations and analysis during immunofluorescence assays.

## In situ hybridization

Advanced Cell Diagnostics (ACD) performed probe design and manufacturing (*Tsai et al., 2019*). In situ hybridization was performed according to manufacturer recommendations based on the RNA-scope 2.5 Chromogenic Assay technology. The single-plex protocol using only the *eya2* probe was performed on 5 dpa blastema sections. The duplex protocol using *eya2* (red) and *prrx1* (blue) probe was used on intact (0 dpa) and 7 (early-bud), 14 (mid-to-late-bud) blastemas and 21 dpa pallete sections. High magnification images were acquired using a Nicon Eclipse Ni fluorescent microscope equipped with a Nikon DS-Ri2 camera. Low magnification images and stitching was performed using a Zeiss fluorescent microscope with an automated platform.

## Immunofluorescence

### For EdU/pH3 double stain

Slides were rehydrated for 15 min with PBS and permeabilized using 0.5% Triton-X-100 PBS. Slides were then rinsed two times with 3% BSA followed by EdU visualization following manufacturers recommendations (Click-iT EdU, Thermo Fisher). Slides were washed with 3% BSA for 5 min followed by PBS. Slides were then incubated with 2% BSA/0.1% Triton-X-100 PBS for 1 hr followed by an overnight incubation at 4°C using 1:400 anti-phospho Histone H3 (Ser10) (#06–570, Millipore) diluted with blocking solution. Slides were washed thrice with PBS followed by secondary antibody

incubation for 2 hr (1:100, Alexa Fluor 488). Slides were then incubated with DAPI (1.4 μmol/L) in PBS and washed thrice before mounting using hydromount (National Diagnostics). Control slides included were: EdU-only stained slide (to test bleaching through the green channel), pH3-only stained slide (to test bleaching through the red channel) and secondary antibody-only stained slide (to test background). To control for the background in the red channel (EdU), the control slide was incubated with only the azide without the other components of the cocktail mix.

### For EdU/pH2AX double stain

We followed the protocol above with the following changes: Slides were incubated with background sniper (Biocare Medical) for 15 min following the completion of the EdU protocol. Slides were rinsed twice with PBS and anti-gamma H2A.X (phospho S139) (1:250, #ab81299, Abcam) was incubated overnight at 4℃ in 2% BSA/0.1% Triton-X-100 PBS.

To detect apoptotic cells using Terminal deoxynucleotidyl transferase dUTP nick end labeling (TUNEL) we used the in situ cell death detection kit following manufacturers recommendations (Sigma-Aldrich).

### To visualize muscle and axon projections

We used Alexa Fluor 594 Phalloidin at 1:40 (Life Technologies) and anti-beta-III tubulin at 1:50 (TuJ-1, R&D Systems). To quantify bone, muscle and axon area, the threshold feature of ImageJ was used to select positive DAPI, Phalloidin and beta-III tubulin areas. Bone was measure using the freehand line feature to circle the bone using the DAPI images. The average area of the radius and ulna was computed per animal and subsequently used for normalization. The DAPI area and whole section area was used to generate relative values of bone, muscle and beta-III tubulin areas. Images were captured using a Nikon Eclipse Ni fluorescent microscope equipped with a Nikon DS-Ri2 camera or an Evos FL Auto two imaging system. High magnification images presented in *Figures 3h* and *4g* were acquired using the 60x lens and stitched together using the microscope's software. To quantify γ-H2AX intensity across the skin, ImageJ's line tool was used, four perpendicular lines were drawn randomly throughout the skin and measurements of the most intense area were plotted. During γ-H2AX quantification in blastema cells, nucleus area was measured based on the DAPI stain using ImageJ. Individual values of nucleus area, γ-H2AX foci number and presence of EdU for each blastema cell were recorded. γ-H2AX foci were divided by nucleus area to provide the relative presence of γ-H2AX foci in each cell. The range of values for this analysis was from 0 to $65*10^6$ (γ -H2AX foci per DAPI area). % Cells were placed on bins as follows: Zero = 0, 0 < Low < $10*10^6$, $10*10^6$ < Med < $20*10^6$, High >$20*10^6$.

## Immunocytochemistry

Axolotl AL1 fibroblast cells were kept at room temperature with modified L-15 media (0.7x, 5% FBS, 2 mM L-Glutamine, 1x Insulin-Transferrin-Selenium, 1x Penicillin-Streptomycin). AL1 cells were plated on glass coverslips and exposed to UV under a G30T8 UV lamp for 15 min. Controls were covered with aluminum foil. Cells were left to recover for 4 hr before fixation with 4% paraformaldehyde for 15mins before blocking and antibody incubations. Antibodies used were: phospho-histone H2A.X (Ser139/Tyr142) (#5438, Cell Signaling Technology), anti-gamma H2A.X (phospho S139) (#ab11174, Abcam), and anti-gamma H2A.X (phospho S139) (#ab81299, Abcam). Images were captured using a Nicon Eclipse Ni fluorescent microscope equipped with a Nikon DS-Ri2 camera.

## RNA extraction, cDNA preparation and quantitative PCR analysis

Tissues from different time points during regeneration (0, 3, 7, 14, and 21 dpa) were extracted using scissors and placed in TRIzol (Thermo Fisher). TRIzol protocol was followed till aqueous phase was removed. RNA was then extracted using RNeasy Mini Kit (QIAGEN). RNA quality was analyzed using Nanodrop 2000 (Thermo Fisher) and samples having 260/280 > 1.8 were selected for further analysis. 500 ng RNA template was used to generate cDNA using High-Capacity cDNA Reverse Transcription Kit (Thermo Fisher). Reactions with the same amount of RNA without the reverse transcriptase were performed as -RT controls to subtract potential genomic contamination. QPCR was performed using iTaq Universal SYBR Green Supermix (Bio-Rad). Expression was determined by comparing the Ct values of the experimental samples to a standard curve generated by known concentrations and

then normalized by samples' *ef1a* expression. Three independent biological samples were used for each time point, run in experimental triplicates. Only two samples passed our quality control and were used for 7 dpa blastema. Band specificity was assessed using a melt-curve analysis.

Primers used:

Eya2_qPCR_F: 5'-TAGCATAAGTCCATCAGCCA-3',
Eya2_qPCR_R: 5'-TCCCCATCTTTAACTGGTGT-3', expected band: 141 bp,
EF1A_qPCR_F: 5'-AACATCGTGGTCATCGGCCAT-3',
EF1A_qPCR_R: 5'-GGAGGTGCCAGTGATCATGTT-3' as previously described (*Sousounis et al., 2014b*).
Brcc3_qPCR_F: 5'-GCGTCTTGTTCTTCCTGGCA-3',
Brcc3_qPCR_R: 5'-CTTCCAATCTGTCCAGGCCC-3', expected band: 147 bp,
Dhx9_qPCR_F: 5'-GCTCAAATACTCTGCGCTGC-3',
Dhx9_qPCR_R: 5'-CTGAAAGTTCCTGGAGCGGT-3', expected band: 163 bp,
Foxm1_qPCR_F: 5'-GACACCAAGCCCGAATCAGA-3',
Foxm1_qPCR_R: 5'-GAGACCACCACCGCCTTAAA-3', expected band: 245 bp,
Parp1_qPCR_F: 5'-GACAGTTGTCAGCGGGAAGA-3',
Parp1_qPCR_R: 5'-ACTGCACAGGAGACATCAGC-3', expected band: 158 bp,
Rnf168_qPCR_F: 5'-CCCTGCAGATGCAGAAGGAA-3',
Rnf168_qPCR_R: 5'-AGGACGGCTCTGTGAGTTTG-3', expected band: 242 bp,
Rad51_qPCR_F: 5'-GGAGAGTTCCGCACTGGAAA-3',
Rad51_qPCR_R: 5'-CAGCAACTGGGTCTGATGGT-3', expected band: 234 bp,
Ung_qPCR_F: 5'-CCGTTCATCCTTCACCCCTC-3',
Ung_qPCR_R: 5'-GTCAATCAATGCCTTGCCCG-3', expected band: 104 bp,
Xrcc6_qPCR_F: 5'-CCTCTTGCAGAAGGCTGTGA-3',
Xrcc6_qPCR_R: 5'-TAAGCAGTGACAGGGGCTTG-3', expected band: 245 bp,
Xpc_qPCR_F: 5'-TGTACTCACCTGCACGTTCC-3',
Xpc_qPCR_R: 5'-TCCGCTTGTTCGATGTTCCA-3', expected band: 182 bp,
Brca1_qPCR_F: 5'- CGGTGGTGACGAACAAGAGA-3',
Brca1_qPCR_R: 5'- GACCCATTTTCTCCCTGCGA-3', expected band: 215 bp,
Ercc2_qPCR_F: 5'- TATCCGGGGGATCAGATGGT-3',
Ercc2_qPCR_R: 5'- TGAACATACCCGTGTCGTGG-3', expected band: 168 bp,
Mms19_qPCR_F: 5'- ATTGCCGAAGCAGGTTCTCA-3',
Mms19_qPCR_R: 5'- GGTCAGGGCAAGGAGTTTGT-3', expected band: 187 bp,
Rif1_qPCR_F: 5'- TTCGGTCCTAGGCTACGTGA-3',
Rif1_qPCR_R: 5'- GGCGAAGTCCTGTTGTGAGA-3', expected band: 145 bp,
Xrcc1_qPCR_F: 5'- CCCCACGCCAGATGATAGAC-3',
Xrcc1_qPCR_R: 5'- CTCTCGTCATCTGTGGAGCC-3', expected band: 172 bp,
Ogg1_qPCR_F: 5'- GACCAGCAAAAACACCCGTC-3',
Ogg1_qPCR_R: 5'- TGCTACGCTGTCTGCCTTAC-3', expected band: 146 bp,
Msh2_qPCR_F: 5'- CCTCCAGACACCCTTGCATT-3',
Msh2_qPCR_R: 5'- CAGCCAATTTGGGCCATGAG-3', expected band: 172 bp
Fanca_qPCR_F: 5'- AGAGAAGCAACCAGGGAACG-3',
Fanca_qPCR_R: 5'- AGCAGACCGGGGAAAGATTG-3', expected band: 218 bp,
H2afx_qPCR_F: 5'- CGACGAAGAGCTGAACAAGC-3',
H2afx_qPCR_R: 5'- GGAGCTCTTCTTGCCTTTGC-3', expected band: 148 bp.

## DNA extraction, PCR and sequencing

Genomic DNA was extracted using DNeasy Blood and Tissue kit (QIAGEN) following manufacturer's instructions. Quality and quantity of DNA was measured using Nanodrop 2000. PCR amplification of the target area was performed using Phusion High-Fidelity DNA Polymerase (NEB) with 240 ng genomic DNA and the following primers: For genotyping *eya2*-edited animals Eya2_Exon3_F: 5'-CCTCACATTGTATCAAGCCC-3' and Eya2_Intron3_R: 5'-GCTGCACATTTCCAAAACAA-3' (expected band 200 bp), for genotyping the presence of *cre* in the genome Cre_F: 5'-ATTTGCCTGCATTACCGGTC-3' and Cre_R: 5'-ATCAACGTTTTCTTTTCGG-3' (expected band 350 bp), and for verifying successful knock-in of the bait plasmid: Eya2_promoter_F: 5'-TGATTGTGTTGAGGAAATCCA-3' and Cre_R, (expected band 2205 bp). For identifying indels, PCR products were cleaned using AMPure XP beads (Beckman Coulter) and were sequenced at the Massachusetts General Hospital

Center for Computational and Integrative Biology DNA Core using their high-throughput CRISPR sequencing pipeline.

## Generating *eya2* mutants and *cre* knock-in

Genomic sequences were initially retrieved from the Sal-site (www.ambystoma.org/). gRNA design was performed on a manually constructed *eya2* locus composed of SuperContig_67476, SuperContig_81564, SuperContig_127712, SuperContig_93, and SuperContig_256362 (*Smith, 2018*). The *eya2* locus can now be also found at AMEXG_0030001381 contig from the latest axolotl assembly (*Nowoshilow et al., 2018*). At the time of collection, we noticed that the SuperContigs contain more faithful sequence representation of the *eya2* gene (identical to the sequence found in the axolotl transcriptome [*Bryant et al., 2017c*]), whereas the AMEXG assembly contains more faithful distances between exons and introns. All CRISPR/Cas9 targetable sequences were identified using the ZiFiT software and were validated using IDT's CRISPR-Cas9 guide RNA design checker software for on-target performance. The gRNAs with the highest performance were then validated in silico for specificity against the axolotl genome and transcriptome using blastn. Based on the gRNA sequence, crRNA was manufactured by IDT. gRNA sequence for generating *eya2* mutant axolotls: 5'-TACCCAACGATGGCCGCTTA-3' (targeting exon 3), and Eya2-P2A-Cre knock-in axolotls: 5'-CCTACGCTCAACCTTCACGA-3' (targeting exon 1). Axolotl egg handling and injection was performed as previously described (*Fei et al., 2017*). CRISPR/Cas9 complex was generated by incubating crRNA (20 uM final concentration) and tracrRNA (20 uM final concentration) for 5 min at 95°C and allowing them to cool to room temperature. Cas9 protein (0.64 μg/μl final concentration) was added and incubated for 10 min at 37°C and cooled to room temperature before injecting the solution in freshly-laid axolotl eggs. Highly-edited $F_0$ *eya2*-targeted animals were raised to adulthood. $F_1$ *eya2*$^{-/-}$ axolotls were generated by intercrossing highly-edited $F_0$ animals. Knock-in of the *cre* recombinase in the *eya2* locus was performed as previously described (*Fei et al., 2017*). Along with the CRISPR/Cas9 targeting *eya2* exon 1, we co-injected a promoter-less pGEM-T-Eya2-P2A-CRE bait plasmid allowing the *cre* to be expressed under the *eya2* promoter following non-homologous end joining (*Figure 2—figure supplement 2*). Successful knock-in was determined by genotyping the 2205 bp fragment spanning from the endogenous *eya2* promoter to the *cre* recombinase coding sequence (see PCR section). For detecting mCherry in successfully knocked-in animals, limbs were placed under a Leica M165 FC fluorescent dissecting microscope and fluorescent and bright field images were taken with a Leica DFC310 FX camera before fixation.

## Phylogenetic tree construction

Eya1, Eya2, Eya3 and Eya4 protein sequences from several different vertebrates were found at the National Center for Biotechnology Information (NCBI) Proteins database. Axolotl Eya1, Eya2, Eya3 and Eya4 protein sequences were translated from the axolotl transcriptome (*Bryant et al., 2017c*). Sequences were input at the EMBL-EBI Clustal Omega software with output format as ClustalW with character counts and default settings. Tree code was then imported to the Interactive Tree of Life software for visualization with display mode unrooted. The sequences used for the analysis were:

For Eya1: NP_000494.2 – *Homo sapiens*, XP_001164492.1 – *Pan troglodytes*, XP_002805419.1 – *Macaca mulatta*, XP_005638096.1 – *Canis lupus familiaris*, XP_002692745.1 – *Bos taurus*, NP_034294.2 – *Mus musculus*, XP_006237801.1 – *Rattus norvegicus*, XP_418290.4 – *Gallus gallus*, NP_571268.1 – *Danio rerio*, XP_004915225.1 – *Xenopus tropicalis*, c1076020_g1_i1 – *Ambystoma mexicanum*. For Eya2: NP_005235.3 – *Homo sapiens*, XP_003317029.1 – *Pan troglodytes*, XP_002808001.1 – *Macaca mulatta*, XP_853061.3 – *Canis lupus familiaris*, NP_001030541.1 – *Bos taurus*, NP_001258891.1 – *Mus musculus*, NP_569111.1 – *Rattus norvegicus*, NP_990246.1 – *Gallus gallus*, NP_001004558.1 – *Danio rerio*, NP_001090820.1 – *Xenopus tropicalis*, c1077764_g3_i1 – *Ambystoma mexicanum*. For Eya3: NP_001981.2 – *Homo sapiens*, XP_001151069.1 – *Pan troglodytes*, XP_001112789.1 – *Macaca mulatta*, XP_003433895.2 – *Canis lupus familiaris*, NP_001193660.1 – *Bos taurus*, NP_997592.1 – *Mus musculus*, NP_001101380.1 – *Rattus norvegicus*, XP_004947753.1 – *Gallus gallus*, NP_001073521.1 – *Danio rerio*, NP_001116491.1 – *Xenopus tropicalis*, c1078903_g1_i1 – *Ambystoma mexicanum*. For Eya4: NP_742103.1 – *Homo sapiens*, XP_001169695.2 – *Pan troglodytes*, XP_002808446.1 – *Macaca mulatta*, XP_005615485.1 – *Canis lupus familiaris*, XP_005210996.1 – *Bos taurus*, NP_034297.2 – *Mus musculus*, NP_001258246.1 –

*Rattus norvegicus*, XP_004935789.1 – *Gallus gallus*, XP_004914647.1 – *Xenopus tropicalis*, E9QGF5 – *Danio rerio*, c1078337_g1_i1 – *Ambystoma mexicanum*.

Similarly, we performed the phylogenetic tree analysis for the axolotl H2AX protein sequence. The sequences used for the analysis were:

NP_002096.1 – *Homo sapiens*, XP_001102040.1 - *Macaca mulatta*, XP_005619742.1 - *Canis lupus familiaris*, NP_001073248.1 - *Bos taurus*, NP_034566.1 - *Mus musculus*, NP_001102761.1 - *Rattus norvegicus*, XP_004947974.1 - *Gallus gallus*, NP_957367.1 - *Danio rerio*, NP_001015968.1 - *Xenopus tropicalis*, AMEXG_0030106036 - *Ambystoma mexicanum*.

## Protein 3D structure modeling

The human Eya2 protein 3D structure was downloaded from the Protein Data Bank ID 3HB1. Swiss – PdbViewer was used to predict Axolotl Eya2 and visualize its 3D structure.

## DNA Damage Response analysis

RNA-seq expression data were acquired from *Stewart et al. (2013)*. Microarray data were acquired from *Voss et al. (2015)*. 190 genes related to DNA damage/repair and their putative function were acquired from *Wood et al. (2001)*; *Wood et al. (2005)*. Our cutoff values were as follows: For the RNA-seq data, up-regulation was defined when at any regeneration timepoint the gene was expressed at least two times more than the intact, down-regulation was defined when at any regeneration timepoint the gene was expressed at least two times less than the intact, mixed regulation was defined when genes were found both up-regulated and down-regulated at different regeneration timepoints. All other genes were defined as no regulation. For the microarray data, up-regulation was defined when genes were shown to be at least more than one standard deviation versus the intact at any timepoint. Down-regulation, mixed regulation and no regulation was similarly defined. Details for this analysis can be found in *Supplementary file 1*. The DDR gene subset analysis was performed using the RNA-seq data. Genes associated with DNA repair were selected based on the Gene Ontology Term 'Positive Regulation of DNA repair'. There were 55 human proteins in this category and we found 39 of those in the axolotl transcriptome. To ensure that the most DNA repair-specific genes were included in the analysis, we excluded nine genes that were not directly associated with the process (CEBPG, EGFR, FGF10, RPS3, TIMELESS, TRIM28) or not expressed in intact tissue (EYA1, FAM168A, TMEM161A). 0, 3, 5, 7, 10, 14, 21, and 28 dpa time points were included in the analysis (*Supplementary file 1*). Relative expression of each time point compared to 0 dpa was computed and then expression of all genes was averaged. All annotations were based on the human proteome.

## Co-expression correlation analysis

RNA-seq expression data were acquired from *Stewart et al. (2013)*. Time points used were 0, 3, 5, 7, 10, 14, 21, and 28 dpa. Expression profiles of each gene were compared to that of *eya2* and Pearson correlation co-efficiency was calculated. $R^2 > 0.8$ was used as a cutoff. Positively or negatively correlated genes were selected based on the slope of their correlation curve. Genes in each group were uploaded to DAVID functional annotation analysis tool (v6.8) and Gene Ontology Enrichment Analysis was performed using GOTERM_DIRECT categories for Cellular Component, Biological Process and Molecular Function. Categories were considered enriched when False Discovery Rate (FDR, most stringent) was less than 0.05 (*Supplementary file 2*).

## RNAseq: tissue collection, preparation and analysis

Axolotls were amputated at mid-stylopod level and intact/blastema tissues were collected at 0, 5 and 14 dpa. Tissues were flash frozen in liquid nitrogen, RNA was extracted using TRIzol Reagent (Thermo Fisher Scientific) and RNA Clean and Concentrator Kit (Zymo Research). RNA purity (260/280 ~ 2.0, 260/230 ~ 2.0) was checked using Nanodrop (Thermo Fisher Scientific). RNA concentration was detected using Qubit Fluorometric Quantitation (Thermo Fisher Scientific). RNA quality was determined using RNA Integrity Number (RIN >9.0) from a Tapestation System (Agilent). Genotypes were determined as described previously. Biological replicates used: *eya2*$^{-/-}$ 14 dpa n = 3, *eya2*$^{-/-}$ 5 dpa n = 4, *eya2*$^{-/-}$ 0 dpa, *eya2*$^{+/-}$ 14 dpa n = 4, *eya2*$^{+/-}$ 5 dpa n = 4, *eya2*$^{+/-}$ 0 dpa n = 4, *eya2*$^{+/+}$ 14 dpa n = 4, *eya2*$^{+/+}$ 5 dpa n = 4, *eya2*$^{+/+}$ 0 dpa n = 3. 200 ng total RNA was used for RNAseq

through the Bauer Core Facility at Harvard University. Library preparation was performed using Kapa mRNA Hyper Prep with Poly-A Selection. Libraries were tested for quality using Tapestation System (Agilent) and Kapa qPCR Library Quantification. Sequencing was performed using NextSeq High with 2 × 75 bp read lengths (Illumina).~11 million paired-end reads per sample were filtered through Trimmomatic v0.39 using the following settings: ILLUMINACLIP:TruSeq3-PE.fa:2:30:10:2:keepBoth-Reads CROP:73 HEADCROP:11 LEADING:3 TRAILING:3 MINLEN:63 (*Bolger et al., 2014*). This code removed nucleotides from the beginning and the end that appeared not to have correct frequencies based on the FastQC report. The process discarded ~5% of the reads. Read alignment was performed using kallisto (*Bray et al., 2016*) with 100 bootstraps and the axolotl transcriptome annotated with human proteins (evalue <1E-4, 60–70% alignment). Differential gene expression analysis was performed using sleuth (*Pimentel et al., 2017*). To determine genes differentially regulated across genotypes we used the following code: so <- sleuth_fit(so,~Genotype + Timepoint, 'full'), so <- sleuth_fit(so,~Timepoint, 'Genotype'), so <- sleuth_lrt(so, 'Genotype', 'full'). We also performed the naïve analysis to determined differentially regulated genes across both Genotype and Timepoints (*Supplementary file 3*). For Gene Ontology Enrichment analysis, genes with q-value <0.05 were input in DAVID as described previously. For Protein Network analysis, genes with q-value <0.01 were input in STRING (*Szklarczyk et al., 2015*). Genes up-regulated or down-regulated in *eya2*$^{-/-}$ tissues were determined by comparing the transcript per million (TPM) values obtained with kallisto. Pseudo-coloring was performed by selecting enriched terms associated with biological processing. When data-mining for specific genes the q-value and the levels of expression were taken into consideration. Genes used for volcano plots are found in *Supplementary file 3*. Raw data can be accessed in the NIH Sequence Read Archive: PRJNA573629.

## Statistical analyses

For complete statistical analysis for each of the comparisons conducted see *Supplementary file 4*. Briefly, t-test was used when two groups were compared, one-way ANOVA was used when more than two groups were compared, and two-way ANOVA was used when more than two groups and two different treatments were compared. Bonferroni corrections for multiple selections were applied when applicable. Kruskal-Wallis test or Mann-Whitney test were also used when Brown-Forsythe and Bartlett's test, or variances were found to be significant when performing ANOVA or unpaired t test (Both parametric and non-parametric test showed similar results and are both presented in *Supplementary file 4*). Throughout the entire study, all quantifications were performed by researchers blinded to the treatment and/or genotype.

## Acknowledgements

Funding for the project was provided by the Sara Elizabeth O'Brien Trust, Bank of America, NA Trustee Postdoctoral Fellowship (K S), the National Eye Institute of the National Institutes of Health under Award Number K99EY029361 (K S), the Paul G Allen Family Foundation (M L, J L W), NIH Office of the Director (1DP2HD087953 to J L W), the Eunice Kennedy Shriver National Institute of Child Health and Human Development (1R01HD095494 to J L W), Harvard Stem Cell Institute Internship Program (J M F), the Howard Hughes Medical Institute Gilliam Fellowship (D M B). We thank members of the Lehoczky, Tabin, Monaghan, Srivastava, McCusker, Levin, and Whited laboratories for discussions. We thank Amy Wagers and Michael Florea for sharing equipment and expertise. The authors declare no conflicts of interest.

## Additional information

### Funding

| Funder | Grant reference number | Author |
|---|---|---|
| Sara Elizabeth O'Brien Trust | Postdoctoral fellowship | Konstantinos Sousounis |
| National Institutes of Health | K99EY029361 | Konstantinos Sousounis |
| Paul G Allen Family Foundation | Allen Discovery Center at Tufts | Michael Levin<br>Jessica L Whited |

| National Institutes of Health | 1DP2HD087953 | Jessica L Whited |
| National Institutes of Health | 1R01HD095494 | Jessica L Whited |
| Harvard Stem Cell Institute | HIP | Jose Martinez Fernandez |
| Howard Hughes Medical Institute | Gilliam Fellowship | Donald M Bryant |

The funders had no role in study design, data collection and interpretation, or the decision to submit the work for publication.

### Author contributions
Konstantinos Sousounis, Conceptualization, Data curation, Formal analysis, Funding acquisition, Investigation, Visualization, Methodology, Writing - original draft, Writing - review and editing; Donald M Bryant, Conceptualization, Data curation, Formal analysis, Investigation, Methodology; Jose Martinez Fernandez, Samuel S Eddy, Jihee Han, Katharine Courtemanche, Formal analysis, Investigation; Stephanie L Tsai, Gregory C Gundberg, Formal analysis, Investigation, Writing - review and editing; Michael Levin, Conceptualization, Funding acquisition; Jessica L Whited, Conceptualization, Resources, Supervision, Funding acquisition, Methodology, Writing - original draft, Project administration, Writing - review and editing

### Author ORCIDs
Stephanie L Tsai (iD) http://orcid.org/0000-0001-7549-3418
Michael Levin (iD) http://orcid.org/0000-0001-7292-8084
Jessica L Whited (iD) https://orcid.org/0000-0002-3709-6515

### Ethics
Animal experimentation: All experiments involving animals were performed according to IACUC protocol #2016N000369 at Brigham and Women's Hospital and #19-02-346 at Harvard University. All surgeries were performed while animals were anesthetized in tricaine. All experiments were planned and executed in manners that minimized animal suffering.

### Decision letter and Author response
Decision letter https://doi.org/10.7554/eLife.51217.sa1
Author response https://doi.org/10.7554/eLife.51217.sa2

## Additional files

### Supplementary files
• Supplementary file 1. DNA damage response analysis. This file contains gene expression data related to DDR from high-throughput studies. DDR Genes: Data related to *Figure 1a* and *Figure 1—figure supplement 1a*. DDR Analysis – *Figure 1b*: Data related to *Figure 1b*. DDR Analysis Eya2-RNAseq: Data related to *Figure 5b*.

• Supplementary file 2. Eya2 co-expression correlation analysis and Gene Ontology enrichment. This file contains data used for the *eya2* co-expression correlation analysis during regeneration presented in *Figure 2d* and *Figure 2—figure supplement 1*. +Correlation Eya2: Gene expression over time, R (*Vitale et al., 2017*) and slope values for genes positively correlated with *eya2*. GO Enrichment +Correlation: Enriched GO Terms from the +Correlation Eya2 dataset. -Correlation Eya2: Gene expression over, $R^2$ and slope values for genes negatively correlated with *eya2*. GO Enrichment -Correlation: Enriched GO Terms from the – Correlation Eya2 dataset.

• Supplementary file 3. RNAseq data with genotype and naïve expression analysis, Gene Ontology enrichment and protein network analysis. This file contains data related to the RNAseq performed in Eya2 mutant axolotls during regeneration. Data are presented in *Figure 5*, and *Figure 5—figure supplement 1*. Expression values are given as TPMs. Expression – qvalue: All expression values, annotation, and qvalues using both our analysis per sample. GO Analysis q < 0.01 down: GO analysis

using genes differentially expressed between *eya2* genotypes. Network Analysis q < 0.01 down: Enriched terms and associated genes found down-regulated in *eya2*$^{-/-}$. Network Analysis q < 0.01 up: Enriched terms and associated genes found up-regulated in *eya2*$^{-/-}$. G1 S Volcano Plot: Genes related to G1/S transition and used in *Figure 5*. G2 M Volcano Plot: Genes related to G2/M transition and used in *Figure 5*.

- Supplementary file 4. Statistical analysis used throughout the paper. This file contains the statistical analysis (statistical tests and all values related to the statistical test including number of samples tested (n)) used for each Figure and panel. Tabs are labeled based on the Figure and Figure Panel they are referring to.

- Transparent reporting form

## Data availability

Raw data can be accessed in the NIH Sequence Read Archive: PRJNA573629.

The following dataset was generated:

| Author(s) | Year | Dataset title | Dataset URL | Database and Identifier |
|---|---|---|---|---|
| Sousounis K, Bryant DM, Martinez Fernandez J, Eddy SS, Tsai SL, Gundberg GC, Han J, Courtemanche K, Levin M, Whited JL | 2019 | Eya2 and DNA damage response during axolotl limb regeneration | https://www.ncbi.nlm.nih.gov/sra/?term=PRJNA573629 | NCBI Sequence Read Archive, PRJNA573629 |

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
