## [Decision Letter]

**Acceptance summary:**

This paper integrates bioinformatic, genetic, cell biology, and molecular biology approaches to convincingly establish a DNA Damage Response (DDR) during salamander limb regeneration. The results of this work reveal an under-appreciated molecular mechanism in regenerative biology that may prove central to understanding the basis of regenerative ability.

**Decision letter after peer review:**

Thank you for resubmitting your work entitled "Eya2 promotes cell cycle progression by regulating DNA damage response during vertebrate limb regeneration" for further consideration at *eLife*. Your revised article has been favorably evaluated by Didier Stainier (Senior Editor) and four reviewers, one of whom is a member of our Board of Reviewing Editors.

The manuscript has been improved but there are quite numerous remaining issues that need to be addressed, as outlined below:

Essential Revisions:

1) The manuscript is difficult to follow in places and this made it difficult to understand the authors interpretations of crucial results. The manuscript would benefit from deeper conceptual development of DDR during development and regeneration, and conceptually and experimentally address the possibility that eya2 may regulate biological and molecular processes beyond DDR. Please see all reviewer comments.

2) To better establish the essentiality of DDR in limb regeneration, there is need to pursue complementary studies of other DDR components. Please see reviewer 2, 3, and 4 comments.

3) The quality and interpretation of the γ-H2AX immunostaining results is controversial and there is need to more rigorously assess *eya2^-/-^* phenotypes during regeneration and development, which will likely require the generation of new knock-out animals and/or experiments using additional pharmaceutical inhibitors. Please see reviewer 2, 3, and 4 comments.

4) The selection of some genes and expression estimates to establish the existence of a DDR during axolotl limb regeneration comes across as biased, and the manuscript does not provide exacting temporal/developmental stage information about DDR within the context of regeneration. Please see reviewer 1 and 3 comments.

We appreciate that addressing the points above is a significant amount of work. If you choose to address these and other comments raised below and submit a revised manuscript, we would be happy to re-examine the work, assuming that new findings do not significantly alter the scope and impact of the original study.

Reviewer #1:

The manuscript by Sousounis et al. reports on a DNA Damage Response (DDR) during axolotl limb regeneration. Very little is known about DDR in highly regenerative organisms and thus the study addresses an outstanding and significant question. In particular, it is important to discover mechanisms that allow the axolotl and other salamanders to precisely regenerate under conditions that would be expected to engender DNA damage.

A strength of the manuscript is the integrative nature of the work, combining bioinformatic analyses with multiple experimental approaches including molecular, developmental, genetic, and RNA-seq. I am mostly satisfied with the results although it would have been better to look at additional post-amputation time points for the developmental studies. The take-home message in the Abstract is clearly written and focuses on the primary results of the study while the manuscript is quite lengthy and a bit convoluted. Some of the descriptive results are in places too confidently ascribed to mechanism, at least to this reviewer that admittedly has little background in DDR biology. Some of this revolves around a key finding, that genetic and pharmacological inhibition of eya2 leads to different DNA damage outcomes. The manuscript suggests that absence of DNA damage during regeneration in eya2 CRISPR morphants, but presence of a DNA damage with acute pharmacological treatment reveals a compensatory mechanism to mitigate genotoxic stress during development. This result is not mentioned explicitly in the Abstract and perhaps it should be toned back a bit in the Discussion so as not to cloud the central findings of the study. Again, little is known about DDR in highly regenerative organisms and the primary findings in this paper provide a conceptual foundation for future work in this area.

The paper can be improved beyond the typos and grammatical mistakes; the logic was difficult to follow at times. My further comments below are meant to improve the manuscript.

1) It would be better to describe the timing of DDR in more exacting terms, and perhaps more precisely as DNA-Double strand repair given the focus on h2afx?

2) It seems likely that the DDR transcriptional response might be a bit more complicated than presented and additional information is needed to understand how genes were selected to establish that a DDR transcriptional response occurs during regeneration. Genes were mined from Stewart et al., 2013 although other datasets could have been used (e.g. Knapp et al., 2013; Voss et al., 2015). Is there general agreement among the datasets? And, it is not clear why time points before 3 DPA were not included in the analysis. It seems that before 3 DPA, the expression of some/many DDR genes decreases below baseline levels. Is this not important to note in describing the DDR transcriptional response, which seems to coincide with an increase in differentially expressed cell cycle genes at 3 DPA, as noted in previous studies. Dose this suggest integration of DDR and cell cycle transcriptional programs? It is difficult to know which genes were selected from Stewart et al., 2013 because there is no way to cross-reference genes in Supplementary file 1 to that study, and it is not clear if the gene expression estimates were obtained from GEO or the axolomics.org website, which seem to report different estimates and more than one transcript for some genes, including eya2. Is the h2afx transcript annotated correctly; maybe present the mRNA and protein sequence for this gene? Supplementary file 1 seems to be missing a gene name in the last row. The gene ontology term that was used to select genes is not a GO term. Is the correct GO term "positive regulation of DNA repair" and can more be written about the prioritization and elimination process for DDR genes; why for example is parp3 not included? It might provide a more powerful figure to show the overlay of all the expression profiles.

3) Can the Abstract state more exactly when the DDR response is activated? Activated upon injury is not very exact, nor is post-amputation, wording used later in the Results section. The qPCR result (but not RNA-seq results) suggest up-regulation of eya2 by 1 dpa, however phosphorylated and non-phosphoryated h2afx were examined at 14 dpa, and it is not clear what blastema stage was used for the comet assays. Throughout the manuscript, there is need to present results with exact temporal and/or regeneration stage specific information. Might the analyses that were performed missed more dynamic changes in the DDR response?

4) Introduction: Maybe state exactly here what phosphorylation of H2afx means in terms of regulating DNA repair? This is stated much later in the manuscript.

5) Introduction final paragraph: Typo should be "maintaining cell fate…."

6) Leucistic axolotls should be referred to as "white”, this is a distinct axolotl mutant/strain.

7) Results first paragraph: Supplementary file 1 is missing a gene name in last row? Why was the Stewart et al. study used for data mining over other studies….are the results consistent across studies? There are hundreds of genes annotated to DDR processes, how were these selected?

8) Results first paragraph: Can it be stated more exactly when DDR was initiated post-amputation?

9) Subsection “Eya2 is widely expressed in early-bud blastema and is associated with DNA repair and cell cycle during regeneration”: Instead of stating that different time points were examined for the transcript/co-expression analysis, can the exact time points be stated instead?

10) Probably more correct to write that other genes presented expression profiles that were similar/correlated to eya2, instead of writing that the expression of other genes was correlated with eya2. Muscle genes are known to decrease in expression during regeneration (perhaps because muscle cells are being removed from the blastema site), which would be the inverse pattern to increasing eya2 during regeneration. This has been reported many times (e.g. Voss et al., 2015). I don't understand what mechanistic connection is trying to be made here?

11) Paragraph two of subsection “Eya2 is widely expressed in early-bud blastema and is associated with DNA repair and cell cycle during regeneration”: comma after compartmentalize.

12) Subsection “Eya2 mutant axolotls have reduced rate of regeneration and slower cell cycle progression”: When exactly is the DDR initiated upon limb amputation? There is a temporal disconnect between the time of amputation and early bud stage.

13) Instead of writing "indicating that the rate of regeneration….." write suggesting, “that the rate of regeneration…." because smaller limbs might trace to a different mechanism.

14) “17% of eya+/- formed skeletons”….Presumably the other 83% would fully pattern their skeletons with more time allowed in the experiment?

15) How do gene expression data "corroborate our previous molecular and histological data showing decreased cell cycle progression and elevated γ-H2AX levels in blastema tissues lacking eya2 activity"? It is not also clear how a rather abstract GO term implicates a mechanism for compensation for loss of Eya2?

16) How do RNA-seq data "corroborated our previous data indicating slower cell cycle progression and DDR persistence while revealed compensatory mechanisms for the loss of Eya2 function" or, "show an obvious compensatory mechanism"?

17) Final paragraph of subsection “Pharmacological inhibition of Eya2 upon amputation impairs regeneration and confirms its role in DDR.”: Not clear what acquired developmentally means. Couldn't it always be operative during development and re-capitulated in salamanders during regeneration?

18) Subsection “DNA damage response is an integral process of tissue regeneration”: Typo, instead of "shown", write "showed that".

19) At some point, should it not be stated that h2afx is a multifunctional protein, in fact it was first studied because it is a transcriptional coactivator?

20) Subsection “Genetic compensation for the loss of Eya2 function”: You write "assumed during development", which to be consistent with earlier phrasing, should be "acquired during development".

21) Materials and methods: see comment 19 pertaining to acquired developmentally.

22) Materials and methods: it would help understand regeneration rate to state the environmental temperature during experimentation.

23) Subsection “DNA Damage Response analysis”: I think the GO term is "Positive Regulation…" not "Positive Relation..". What database was used as a reference for the GO analysis…..human? Also, how many total genes are in the annotation?

24) Figure 1 legend. Please explain what "relative" means.

25) Figure 1 legend. Typo in this sentence and the conclusion should be toned down. The data are suggestive at best "These data indicate that there Eya2 is associated with DNA repair and cell cycle during limb regeneration".

26) Figure 2—figure supplement 1 legend: Probably should rephrase: "Note that eya2 expression is associated with genes known to participate in cell cycle progression (blue) and DNA repair (red)." What the result suggests is that the eya2 expression profile is correlated with expression profiles for genes that annotate to cell cycle and DNA repair GO terms.

27) Muscle genes are known to decrease in expression during regeneration, independent of eya2 expression during muscle development. This has been reported several times. Again, it is not clear what point is being made.

28) Figure 5—figure supplement 1 legend: I don't see how an inclusive GO term implicates a compensatory mechanism. Regulation of Biological Quality could mean many different things. Here is the definition of this GO term: Any process that modulates a qualitative or quantitative trait of a biological quality. Hundreds of traits come to mind.

Reviewer #2:

In this manuscript, Sousounis et al. investigate the role of the DNA damage response (DDR) during limb regeneration in salamanders. They find the expression of genes associated with the DDR is elevated early on and throughout the process of regeneration. The authors functionally test the role of the DDR by disrupting a member of the eyes absent (EYA) family of proteins. Specifically, they generated eya2 mutant axolotls and used phosphorylation of the H2AX histone variant as a readout of the DDR activation. They also used chemical inhibition of eya2 and in some cases recapitulated the effects observed with the genetic manipulations. As presented, however, the work needs to be strengthened and may benefit by incorporating additional experiments to support the central claim the authors are making. Below, are some suggestions the authors may want to consider.

1) The focus on DDR during regeneration is commendable, but the analysis is perceived as a little premature. Eya2 may function as a transcriptional activator and as a protein phosphatase that regulates many pathways including repair of DNA double-strand breaks. The genotyping of the mutants suggests both the phosphatase and transactivating domains were removed, but the characterization of eya2 was only focused on DNA repair without considering other potential roles.

2) The work relies on the γ-H2AX signal, but the immunostainings in Figure 2E-G are challenging to interpret. For example, in Figure 2E, γ-H2AX appears restricted to the border of the tissue, perhaps labeling epithelial cells. This marginal signal appears to increase in *eya2^-/-^,* but there is not a satisfactory explanation about the spatial distribution and the way the signal looks. According to the authors, the DDR is elevated in proliferating cells, but the distribution of γ-H2AX in the periphery probably implies that other cells also overexpress the signal? Also, the images presented in Figure 2G are consistent with γ-H2AX foci, but the intensity is very low (compare to Figure 1—figure supplement 1C). On the same figure, the EdU stain also looks dim, and it is troublesome to make any interpretations based on the overlapping images in Figure 2H-I. The staining protocol may need some optimization, and there should be some satisfactory explanation about the spatial distribution of the signal and the potential identity of cells with increased γ-H2AX in the epithelia.

3) Intriguingly, *eya2^-/-^*has a mild effect on regeneration compared with the effects of the inhibitor treatment. Also, how regeneration was analyzed is not consistent between the two treatments. For example, in Figure 1E, the authors quantified Zeugopod length over animal length, while in Figure 3H, the analysis was based on relative blastema length. Please clarify why the difference between these parameters. Perhaps standardizing the measurement of the blastema surface area in both cases would be more accurate.

4) The quantification of cell death in *eya2^-/-^*is missing. The Figure 4—figure supplement 1 shows some TUNEL/ γ-H2AX staining, but it is difficult to discern what part is related to background and/or positive signal. Likewise, the amount of TUNEL + cells in the control group Figure 4I, K (below 1%) seems very low considering the DDR is activated, and regeneration requires cell death. This brings another issue related to the amount of DNA damage induced by the treatment with DMSO (Figure 4N). It seems DMSO-intact/blastema groups show a large number of cells with COMET score 3-4. If the Eya2 inhibitor is inducing DNA damage, this will account only for the slight increase of cells with COMET score four. Validating cell death with other approaches (e.g., caspase, FACS, etc.), and adding better contrast or some validation of the staining involving TUNEL may benefit the work.

5) The model presented in Figure 5 is not intuitive, and some of the statements lack support. For example, no data is supporting the presence of replication errors. The cell cycle arrest may need additional experiments, perhaps cell cycle with FACS or other methods. Using EdU and H3P staining to deduce cell cycle arrest is probably a good start, but additional experiments are needed to support this conclusion.

6) The authors may want to consider expanding their characterization of factors involved in the DDR and DNA repair. The DDR influences cell fate in the presence of DNA damage, so incorporating characterization of other molecules such as p53 may strengthen the case.

Reviewer #3:

During regeneration and tissue repair, cells must balance the requirement for highly regulated, brisk proliferation with the requirement to avoid DNA damage or transformation that might occur downstream of replicative stress. Stem cells, in vivo, are thus expected to possess an enhanced ability to proliferate without acquiring DNA damage. Understanding how stem and progenitor cells avoid DNA damage despite rapid cell division, especially in the context of a highly regenerative organism, could improve approaches directed at deriving and utilizing stem cells for regenerative therapies.

It is within this knowledge gap that Sousounis and colleagues begin their studies. Using axolotls, the group created animals with mutations in eyes absent 2, a salamander gene homologous to a group of genes that encode dual-function proteins (transcriptional activator and phosphatase) with roles in morphogenesis, DNA damage repair, and proliferative control. Using these mutants, chemical inhibitors of Eya and other approaches directed at understanding DNA damage during vertebrate limb regeneration, the group presents data to support the idea that DNA damage response (DDR) pathways are important during regeneration. Sousounis, et al., ask an important question about the ways that organisms prevent DNA damage during regeneration, but the overall conclusions of the manuscript lack support in several key areas. Suggestions for required and recommended changes for the work follow. Major revisions and additional experiments would be required for publication in *eLife*.

Major critiques:

1) Eyes absent homologs play roles in DNA repair, but also regulate morphogenesis of several tissues (e.g. eyes, kidneys, muscle, ears, heart) though protein-protein interactions with transcriptional regulators like Six and Dachsous family members. The fact that *eya2^-/-^*does not cause increased DNA damage further complicates the authors' interpretation that DDR is the key function of Eya2. Thus, the claim that Eya2's function in axolotl limb regeneration is due to DNA damage response (and not general transcriptional dysregulation) is undersupported.

a) The authors should clarify whether axolotl Eya2 plays any roles in development. The authors say that limbs are normal in *eya2^-/-^*mutants, but provide only one image per genotype with no scale bars. Is the skeleton normal (e.g. zeugopod length) in uninjured limbs? The authors indicate that there is an intra-peritoneal edema in mutants, but do not discuss the etiology of this phenotype. Overall, a more comprehensive analysis of the knockout phenotype (before injury) is needed. If Eya2 plays roles in axolotl development or physiology, the authors should discuss whether other roles for eya2 might alter interpretation of their regeneration results. This might also determine whether a reader would conclude that the role of Eya2 is regeneration-specific.

b) The authors themselves show considerable transcriptional dysregulation after eye2 knockout (Figure 4), which further highlights other potential roles of Eya2 beyond the DDR. The markers used in Figure 4E do not seem comprehensive. Would it be worth looking at markers of muscle or neurons in the regenerating limb given other developmental roles for eya homologs?

c) The authors should also demonstrate that perturbation of other DNA damage repair mechanisms causes similar limb regeneration failure. This should be achievable by chemical inhibition of canonical DNA damage repair machinery (e.g. inhibitors of ATM, ATR, Chk1, or Chk2).

2) In several sections, the authors have quantified western blot images that are scanned from film that was developed after ECL. It is very challenging to quantify proteins accurately with this approach, due to issues around linearity of signal in both the ECL step and the development step. A great deal of caution should be exercised with these approaches and the conclusions drawn from them. If the authors wish to make quantitative statements about these western blots, more controls need to be completed to ensure that blots were in the linear range and that loading was consistent. Alternatively, another approach (e.g. fluorescent secondary antibodies rather than chemiluminescence) should be considered. Overloading or saturation looks especially likely in Figure 3A and Figure 4L which are key pieces of data for the authors' argument that Eya2 is important for DDR.

3) Validation of the H2AX antibody (and phosphospecific antibodies) seems important for the arguments put forward in this manuscript. It would be helpful to show that the authors see an increase in H2AX phosphorylation (with their antibodies and western blotting/immunostaining) after UV exposure in axolotls. This would validate the antibodies for use in this species and confirm that H2AX phosphorylation behaves as expected in axolotls. The authors should also demonstrate that the γ-H2AX staining (Figure 3E, Figure 4—figure supplement 1A?) reveals nuclear localization, as would be expected for a histone. In some images, the staining looks quite clearly cytoplasmic, though the images are not very high magnification.

Reviewer #4:

Although this manuscript presents new knowledge with the first demonstration of a DNA damage response (DDR) during regeneration, I am not sure if this alone is enough to justify publication in *eLife*. I admit that having read the manuscript several times, I find it difficult to glean what is known and what is unknown about DDR and Eya2 in other systems. Thus, the readability and focus of the manuscript are not helping in its evaluation. From what I gather from other systems, DDR is a common feature of rapidly proliferating cells and can be a consequence of not only acute genotoxic insults but also regular replicative stress that accompanies a rapid S phase. This may be particularly true in axolotls, which must replicate over 30Gb of genome and yet can still accomplish a 48-72hour cell cycle during regeneration. If this is true, I do not find the results to highlight any regeneration-specific insights or elucidate a pressing question concerning how regeneration occurs. For me, there are two potential follow up questions that would address this.

The authors do not report any effect of Eya2 mutation on the development of the organism. First, is it slower than the development of the wildtype animal? Do the authors also observe the same H2AX foci observed during regeneration? If they could show that the DDR process they observe and Eya2's specific role is unique to regeneration and does not have any role in the rapid proliferation and development of the embryo, then I think this would be quite interesting and would potentially be a mechanism where regeneration and development diverge.

Secondly, the mechanism of Eya2 seems to be in the regulation of the DDR such that Eya2's phosphatase activity removes cell cycle checkpoints and indicates a successful resolution of the DNA damage. This seems to be a known role of Eya2 in other animal systems. Thus, although the authors observe phenotypes on cell cycle progression and regeneration, all DNA damage has been successfully repaired (as indicated by a normal comet assay). I think it would be more interesting to determine what happens when DNA damage cannot be repaired. Are there gross effects when progenitors accumulate unrepaired damage sites? Because regeneration can occur multiple times, does accumulated DNA damage impact subsequent regeneration events? This might be achieved through pharmacological inhibition of ATM which is the kinase that phosphorylates H2AX and marks DNA damage sites. Again, this would be unique to regeneration and be of interest to the regeneration field as well as the general stem cell field to look at the consequences of genotoxic stress on progenitor exhaustion.

---

## [Author Response]

Essential Revisions:1) The manuscript is difficult to follow in places and this made it difficult to understand the authors interpretations of crucial results. The manuscript would benefit from deeper conceptual development of DDR during development and regeneration, and conceptually and experimentally address the possibility that eya2 may regulate biological and molecular processes beyond DDR. Please see all reviewer comments.2) To better establish the essentiality of DDR in limb regeneration, there is need to pursue complementary studies of other DDR components. Please see reviewer 2, 3, and 4 comments.3) The quality and interpretation of the γ-H2AX immunostaining results is controversial and there is need to more rigorously assess eya2^-/-^ phenotypes during regeneration and development, which will likely require the generation of new knock-out animals and/or experiments using additional pharmaceutical inhibitors. Please see reviewer 2, 3, and 4 comments.4) The selection of some genes and expression estimates to establish the existence of a DDR during axolotl limb regeneration comes across as biased, and the manuscript does not provide exacting temporal/developmental stage information about DDR within the context of regeneration. Please see reviewer 1 and 3 comments.We appreciate that addressing the points above is a significant amount of work. If you choose to address these and other comments raised below and submit a revised manuscript, we would be happy to re-examine the work, assuming that new findings do not significantly alter the scope and impact of the original study.

In summary these are the areas where we provide additional experiments:

– DNA damage response: We now also analyze and present data from a published microarray timecourse experiment (in addition to the RNAseq data). We have now validated genes using qPCR.

– H2AX regulation: We verified orthology through phylogenetic tree analysis, verified sequence/function via comparative alignment and verified expression using qPCR. We performed western blot analysis of the H2AX status which we could normalize non-histone controls like GAPDH.

– For *eya2*^-/-^ mutant axolotls: We performed extensive characterization of limbs during development with measurements and quantifications of their length as well as bone, muscle and nerves. We also performed a comprehensive histological examination of their organs and tissues throughout their entire body.

– For claims regarding G1/S and G2/M checkpoint activation: We are now using western blot for cell cycle regulators that confirm and validate our EdU/pH3 data indicating G1/S and G2/M stalling of the cell cycle in absence of Eya2 activity.

– For experiments regarding the investigation of additional DDR components: We now provide data that upon inhibition of the DNA damage checkpoint kinases Chk1 and Chk2, limb regeneration is severely impaired.

– We have made changes on how the manuscript reads and how the figures are presented. The paper now has 7 main figures to illustrate the main points of the experiments.

Specific comments/changes for each of the comments by the reviewers are provided below:

Reviewer #1:The manuscript by Sousounis et al. reports on a DNA Damage Response (DDR) during axolotl limb regeneration. Very little is known about DDR in highly regenerative organisms and thus the study addresses an outstanding and significant question. In particular, it is important to discover mechanisms that allow the axolotl and other salamanders to precisely regenerate under conditions that would be expected to engender DNA damage.A strength of the manuscript is the integrative nature of the work, combining bioinformatic analyses with multiple experimental approaches including molecular, developmental, genetic, and RNA-seq. I am mostly satisfied with the results although it would have been better to look at additional post-amputation time points for the developmental studies. The take-home message in the Abstract is clearly written and focuses on the primary results of the study while the manuscript is quite lengthy and a bit convoluted. Some of the descriptive results are in places too confidently ascribed to mechanism, at least to this reviewer that admittedly has little background in DDR biology. Some of this revolves around a key finding, that genetic and pharmacological inhibition of eya2 leads to different DNA damage outcomes. The manuscript suggests that absence of DNA damage during regeneration in eya2 CRISPR morphants, but presence of a DNA damage with acute pharmacological treatment reveals a compensatory mechanism to mitigate genotoxic stress during development. This result is not mentioned explicitly in the Abstract and perhaps it should be toned back a bit in the Discussion so as not to cloud the central findings of the study. Again, little is known about DDR in highly regenerative organisms and the primary findings in this paper provide a conceptual foundation for future work in this area.

We have now toned down our interpretation of this data in the Results and Discussion. We have highlighted instances of compensation in axolotls and other models that have been targeted using CRISPR/Cas9. Since this compensatory mechanism appears to be common with other models. Since this is not the main point of the paper, we do not favor including mention of it in the Abstract, and nonetheless, given the word limit, there is not space.

The paper can be improved beyond the typos and grammatical mistakes; the logic was difficult to follow at times. My further comments below are meant to improve the manuscript.1) It would be better to describe the timing of DDR in more exacting terms, and perhaps more precisely as DNA-Double strand repair given the focus on h2afx?

We now describe that DDR is evident in early-bud blastema and peaks at late-bud blastema. In our field, it is important to define timing in terms of these kinds of anatomical markers rather than specific time points post-regeneration because animals hit these markers at different rates depending on size.

We do not favor defining the role of h2afx in this context so specifically to be only double-strand break repair because we have so far only implicated it in genomic integrity, i.e., the assay will not specifically identify the cause of compromised genomic integrity as being double-strand breaks.

2) It seems likely that the DDR transcriptional response might be a bit more complicated than presented and additional information is needed to understand how genes were selected to establish that a DDR transcriptional response occurs during regeneration. Genes were mined from Stewart et al., 2013 although other datasets could have been used (e.g. Knapp et al., 2013; Voss et al., 2015).

We now provide data from Voss et al., 2015. In addition to our initial analysis, we have now included a more exhaustive list of genes which we specifically analyzed from these datasets, and the total number of those we investigated is 190.

Is there general agreement among the datasets?

Yes there is general agreement, and this data is now presented in Figure 1A.

And, it is not clear why time points before 3 DPA were not included in the analysis. It seems that before 3 DPA, the expression of some/many DDR genes decreases below baseline levels. Is this not important to note in describing the DDR transcriptional response, which seems to coincide with an increase in differentially expressed cell cycle genes at 3 DPA, as noted in previous studies. Dose this suggest integration of DDR and cell cycle transcriptional programs?

Certainly, our data throughout the paper point toward the direction that DDR is elevated when cell cycle is active. We now provide expression data from all timepoints in Figure 1—figure supplement 1A. While there are some genes whose expression does decrease very early following amputation, the majority of them actually go up, and this relationship, combined with the rest of our data focusing on the later time points, makes us hesitant to draw too many conclusions from this observation. We agree that a future study could specifically investigate the possible coupling of the expression of genes to cell-cycle re-entry early in the regeneration process. Also, because there is probably less concordance between the behavior of individual cells in the very beginning of the process (for example, some will die while others will go on to proliferate), it is possible these early time points exhibit more variability in gene expression across the DDR

response.

It is difficult to know which genes were selected from Stewart et al., 2013 because there is no way to cross-reference genes in Supplementary file 1 to that study, and it is not clear if the gene expression estimates were obtained from GEO or the axolomics.org website, which seem to report different estimates and more than one transcript for some genes, including eya2.

The data were provided from the supplementary files from the original publication (Stewart et al., 2013), which includes a file that has a single expression value per gene.

Is the h2afx transcript annotated correctly; maybe present the mRNA and protein sequence for this gene?

It is important to note that the sequence of h2afx that we focus on in this work was manually mined out of the published axolotl genome paper (see Materials and methods). We present the translated protein sequence of this axolotl h2afx gene; we verify putative functions via multiple sequence alignment; and we verify its expression using qPCR.

Supplementary file 1 seems to be missing a gene name in the last row.

Thank you for noticing that. The last row contained the average values which ultimately were used to plot the average DDR; it was not another gene row with a missing gene name. We have now properly labeled the last row.

The gene ontology term that was used to select genes is not a GO term. Is the correct GO term "positive regulation of DNA repair" and can more be written about the prioritization and elimination process for DDR genes; why for example is parp3 not included? It might provide a more powerful figure to show the overlay of all the expression profiles.

Thank you for noticing this. In the original submission, in the Materials and methods only (not in the main text), the GO term was mislabeled as “positive relation of DNA repair,” which indeed is not a GO term and makes no grammatical sense. This was a typo. We have now properly labeled the GO term.

We are now more explicit in the elimination process of genes in this subcategory. Finally, we have also now included a figure that has an exhaustive list of 190 DNA damage/repair genes and their expression. This analysis is overlaid with their functional subcategories in Figure 1—figure supplement 1A.

3) Can the Abstract state more exactly when the DDR response is activated? Activated upon injury is not very exact, nor is post-amputation, wording used later in the Results section.

We are now more precise in stating the DDR timing. We now state that, “Here we found an innate DNA damage response mechanism that is evident during blastema proliferation (early- to late-bud)...”

The qPCR result (but not RNA-seq results) suggest up-regulation of eya2 by 1 dpa, however phosphorylated and non-phosphoryated h2afx were examined at 14 dpa, and it is not clear what blastema stage was used for the comet assays. Throughout the manuscript, there is need to present results with exact temporal and/or regeneration stage specific information. Might the analyses that were performed missed more dynamic changes in the DDR response?

We are now more precise in stating the DDR timing and the timepoints at which the different experiments were performed. We also now provide H2AX protein phosphorylation regulation at multiple time points. Throughout the manuscript, we are now more careful to report timing, and we say “early-bud,” “mid-bud,” etc. where appropriate, and we also include time points.

4) Introduction: Maybe state exactly here what phosphorylation of H2afx means in terms of regulating DNA repair? This is stated much later in the manuscript.

We have now thoroughly explained this in the first paragraph of the Introduction.

5) Introduction final paragraph: Typo should be "maintaining cell fate…."

We have now fixed this.

6) Leucistic axolotls should be referred to as "white”, this is a distinct axolotl mutant/strain.

We have now fixed this.

7) Results first paragraph: Supplementary file 1 is missing a gene name in last row? Why was the Stewart et al. study used for data mining over other studies….are the results consistent across studies? There are hundreds of genes annotated to DDR processes, how were these selected?

Please see our previous responses for addressing these issues.

8) Results first paragraph: Can it be stated more exactly when DDR was initiated post-amputation?

Our data clearly detect a DDR at the early-bud blastema, and it peaks at mid-bud

blastema. We have now made these points clear in the text.

9) Subsection “Eya2 is widely expressed in early-bud blastema and is associated with DNA repair and cell cycle during regeneration”: Instead of stating that different time points were examined for the transcript/co-expression analysis, can the exact time points be stated instead?

We have now mentioned the exact time points.

10) Probably more correct to write that other genes presented expression profiles that were similar/correlated to eya2, instead of writing that the expression of other genes was correlated with eya2. Muscle genes are known to decrease in expression during regeneration (perhaps because muscle cells are being removed from the blastema site), which would be the inverse pattern to increasing eya2 during regeneration. This has been reported many times (e.g. Voss et al., 2015). I don't understand what mechanistic connection is trying to be made here?

We made the recommended change. We did not mean to make a point about the whole suite of muscle differentiation genes going down at the tip of the stump during early regeneration; this point has indeed been made in several published papers, and it matches the architectural disassembly of muscle fibers that has been appreciated for decades at the histological level.

Instead, we were trying to highlight a correlation between an increase in the transcript under study, *eya2*, and the activation of muscle progenitor cells known occur in this time frame, as well as the diminishment of mature muscle transcripts. Put another way, the correlation analysis simply shows that when there is high *eya2* expression, there is low expression of differentiated muscle markers. Similarly, when there is high expression of differentiated muscle markers, there is low expression of *eya2*. While this is just a correlation, it fits with a model in which differentiated muscle does not express a lot of *eya2*, while muscle progenitor cells do (which is exactly what we see in the RNA *in situ* hybridizations).

11) Paragraph two of subsection “Eya2 is widely expressed in early-bud blastema and is associated with DNA repair and cell cycle during regeneration”: comma after compartmentalize.

Change was made as recommended.

12) Subsection “Eya2 mutant axolotls have reduced rate of regeneration and slower cell cycle progression”: When exactly is the DDR initiated upon limb amputation? There is a temporal disconnect between the time of amputation and early bud stage.

We now provide the gene expression data from all timepoints available. DDR is evident in early-bud blastema (~5 dpa) and peaks at late-bud blastema (~14 dpa). It is not clear how DDR changes between 0 and 5 dpa. As we speculated earlier in this response document, this is probably because of heterogeneity in cell behaviors soon after amputation (for example, there is tissue remodeling with many cells triggering the apoptotic program in addition to progenitor cells starting to enter the cell cycle). Thus, between 0-5 dpa there are multiple events that contribute to the DDR pathway differently at the cellular level.

13) Instead of writing "indicating that the rate of regeneration….." write suggesting, that “the rate of regeneration…." because smaller limbs might trace to a different mechanism.

Change was made as recommended.

14) “17% of eya+/- formed skeletons”….Presumably the other 83% would fully pattern their skeletons with more time allowed in the experiment?

Yes, we made have now more clearly made this point. We say, “At 30 dpa, a preselected time point which captures regeneration before the end of the growth process, we collected the regenerated limbs... “

Later in the same paragraph, we note that the mutants do ultimately regenerate limbs, but they do so at a reduced rate.

15) How do gene expression data "corroborate our previous molecular and histological data showing decreased cell cycle progression and elevated γ-H2AX levels in blastema tissues lacking eya2 activity"? It is not also clear how a rather abstract GO term implicates a mechanism for compensation for loss of Eya2?

Changes were made to make these points clearer.

We now explicitly explain that these data show there is an elevated DDR. This now reads, “These data indicate an elevated DNA damage response in *eya2*^-/-^ mutant blastema cells, which corroborates our previous molecular and histological data showing decreased cell cycle progression and elevated γ-H2AX levels (Figure 4H).”

With respect to the other point, we agree that original submission was quite vague about this point. We now say, “On the other hand, the most over-represented terms on the dataset with genes up-regulated in *eya2*^-/-^ tissues were regulation of biological quality, a GO term that encapsulates genes related to driving cellular function to homeostasis (Figure 5—figure supplement 1C, red nodes and Supplementary file 3) which may be part of a mechanism that compensates for the loss of Eya2 to alleviate the elevated DDR and stress.” All of the other GO terms were very specific, but these accounted for very few of the genes. This GO term was positive and also accounted for nearly all of the genes whose expression was significantly changed in the mutants and detectable on RNAseq.

16) How do RNA-seq data "corroborated our previous data indicating slower cell cycle progression and DDR persistence while revealed compensatory mechanisms for the loss of Eya2 function" or, "show an obvious compensatory mechanism"?

Changes were made to more clearly make these point, to be more specific, and to “tone down” the wording:

We now say, “Taken together, RNAseq analysis revealed down-regulation of genes involved in cell cycle progression specifically involved at the G1/S and G2/M checkpoints as well as H2AX deregulation which corroborated our previous molecular data showing reduced EdU/pH3 staining (Figure 3K) while H2AX phosphorylation was aberrantly regulated in *eya2* mutant blastema cells (Figure 4A). In addition, the RNAseq data revealed up-regulation of genes consistent with an active compensatory mechanism to alleviate stress and the loss of Eya2 which included the up-regulation of the *eya2* gene itself.”

Now instead of saying, “obvious,” we say, “potential.”

17) Final paragraph of subsection “Pharmacological inhibition of Eya2 upon amputation impairs regeneration and confirms its role in DDR.”: Not clear what acquired developmentally means. Couldn't it always be operative during development and re-capitulated in salamanders during regeneration?

We have toned down all phrases that included “acquired developmentally”. We now focus/redirect the conversation for this compensatory mechanism to papers that study it in more detail.

18) Subsection “DNA damage response is an integral process of tissue regeneration”: Typo, instead of "shown", write "showed that".

We have fixed this.

19) At some point, should it not be stated that h2afx is a multifunctional protein, in fact it was first studied because it is a transcriptional coactivator?

This is an interesting thing to consider (a possible role as a transcriptional coactivator in regeneration for either H2AX and/or Eya2). However, embarking on these questions is outside the scope of this current work. Given the length of the current manuscript, we believe it is better to focus on the roles we have examined here.

20) Subsection “Genetic compensation for the loss of Eya2 function”: You write "assumed during development", which to be consistent with earlier phrasing, should be "acquired during development".

We have fixed this.

21) Materials and methods: see comment 19 pertaining to acquired developmentally.

We have fixed this.

22) Materials and methods: it would help understand regeneration rate to state the environmental temperature during experimentation.

We now note the temperature in the Materials and methods.

23) Subsection “DNA Damage Response analysis”: I think the GO term is "Positive Regulation…" not "Positive Relation..". What database was used as a reference for the GO analysis…..human? Also, how many total genes are in the annotation?

We have now fixed this, and we have noted the total number of proteins in the Materials and methods.

24) Figure 1 legend. Please explain what "relative" means.

We have fixed this (relative to intact).

25) Figure 1 legend. Typo in this sentence and the conclusion should be toned down. The data are suggestive at best, "These data indicate that there Eya2 is associated with DNA repair and cell cycle during limb regeneration".

Note that this is now Figure 2. We toned it down by adding the word “suggest.” It now reads, “These data suggest that Eya2 is associated with DNA repair and cell cycle during limb regeneration.”

26) Figure 2—figure supplement 1 legend: Probably should rephrase: "Note that eya2 expression is associated with genes known to participate in cell cycle progression (blue) and DNA repair (red)." What the result suggests is that the eya2 expression profile is correlated with expression profiles for genes that annotate to cell cycle and DNA repair GO terms.

We have now rephrased this as suggested.

27) Muscle genes are known to decrease in expression during regeneration, independent of eya2 expression during muscle development. This has been reported several times. Again, it is not clear what point is being made.

We have addressed this in the previous responses about the point we were trying to make.

28) Figure 5—figure supplement 1 legend: I don't see how an inclusive GO term implicates a compensatory mechanism. Regulation of Biological Quality could mean many different things. Here is the definition of this GO term: Any process that modulates a qualitative or quantitative trait of a biological quality. Hundreds of traits come to mind.

Please see our previous comments on this. As we mentioned earlier, this GO term best encompasses the overwhelming majority of the genes that we recovered from the analysis.

Reviewer #2:In this manuscript, Sousounis et al. investigate the role of the DNA damage response (DDR) during limb regeneration in salamanders. They find the expression of genes associated with the DDR is elevated early on and throughout the process of regeneration. The authors functionally test the role of the DDR by disrupting a member of the eyes absent (EYA) family of proteins. Specifically, they generated eya2 mutant axolotls and used phosphorylation of the H2AX histone variant as a readout of the DDR activation. They also used chemical inhibition of eya2 and in some cases recapitulated the effects observed with the genetic manipulations. As presented, however, the work needs to be strengthened and may benefit by incorporating additional experiments to support the central claim the authors are making. Below, are some suggestions the authors may want to consider.1) The focus on DDR during regeneration is commendable, but the analysis is perceived as a little premature. Eya2 may function as a transcriptional activator and as a protein phosphatase that regulates many pathways including repair of DNA double-strand breaks. The genotyping of the mutants suggests both the phosphatase and transactivating domains were removed, but the characterization of eya2 was only focused on DNA repair without considering other potential roles.

In our revised manuscript, we have done considerably more work to address the role of DDR during limb regeneration beyond just implicating Eya2. This work sums to one full figure (half of Figure 1 and half of Figure 6). For example, we now present analyses of the DDR itself during regeneration, before we zero in on Eya2. We also chose an independent inhibitor of the DDR, Chk1/2 inhibitor, and we tested its effects postamputation. We discovered that inhibition of Chk1/2 dramatically impairs axolotl limb regeneration.

In the original submission in the Introduction, we did mention that Eya2 has two separate identified functions in other systems: transcriptional co-activation and phosphatase activity. So far, we have collected evidence that Eya2 is important for a DDR in axolotl limb regeneration, so that is what we chose to pursue in this work. We have not ruled out any potential role for it as a possible transcriptional co-activator in this system/process. Indeed, that would be quite interesting to investigate, but we feel it is outside the scope of this current study, especially since new tools would need to be generated to test this discrete possibility. In the revised Conclusion, we have now speculated that it will be important in the future to also consider a possible transcriptional co-activation role for Eya2 during limb regeneration. We say, “As Eya proteins have been shown to act as transcriptional co-activators in other systems, considering a possible co-activation role for Eya2 in axolotl limb regeneration will also be important and the subject of future study.”

2) The work relies on the γ-H2AX signal, but the immunostainings in Figure 2E-G are challenging to interpret. For example, in Figure 2E, γ-H2AX appears restricted to the border of the tissue, perhaps labeling epithelial cells. This marginal signal appears to increase in eya2^-/-^, but there is not a satisfactory explanation about the spatial distribution and the way the signal looks. According to the authors, the DDR is elevated in proliferating cells, but the distribution of γ-H2AX in the periphery probably implies that other cells also overexpress the signal? Also, the images presented in Figure 2G are consistent with γ -H2AX foci, but the intensity is very low (compare to Figure 1—figure supplement 1C). On the same figure, the EdU stain also looks dim, and it is troublesome to make any interpretations based on the overlapping images in Figure 2H-I. The staining protocol may need some optimization, and there should be some satisfactory explanation about the spatial distribution of the signal and the potential identity of cells with increased γ-H2AX in the epithelia.

We make this clearer and more distinct in the text. There is robust staining in the epithelium that covers all cell layers (see Figure 4 —figure supplement 1). This staining pattern prohibited us from looking for γ-H2AX in the nucleus of the epithelia, thus, excluding them from further analysis. We do think the staining in the cytoplasm of skin cells is likely legitimate, but we do not know why γ-H2AX protein is localizing there. Note that we fixed the limb, embedded it, and cryosectioned it before subjecting it to immunohistochemistry.

For the staining in blastema cells: We have extensively optimized the protocol for γ- H2AX in tissue sections and it still does suffer from background, which is often an issue in axolotl limb immunohistochemistry. This is the first time, to our knowledge, that g-H2AX immunohistochemistry has been presented for axolotl tissues. The antibody is usually used on cultured cells, where autofluoresence is usually less of an issue. Please note that in our supplementary figure, where we use the antibody on axolotl cells in culture, the staining is much cleaner.

Nonetheless, all the foci were quantified by blind (to the experimental design and genotype) researchers and it is consistent with the differences seen between EdU+ and EdU- cells (more γ-H2AX in EdU+ irrespective of genotype). The images presented in Figure 1—figure supplement 1C are from in vitro cultured cells. There are instances of γ-H2AX staining outside the nucleus but we could not correlate the spatial pattern to any function found in the literature. In addition, we do not know if it is due to antibody specificity issues even though the same phenomenon was obtained using two different antibodies.

3) Intriguingly, eya2^-/-^ has a mild effect on regeneration compared with the effects of the inhibitor treatment. Also, how regeneration was analyzed is not consistent between the two treatments. For example, in Figure 1E, the authors quantified Zeugopod length over animal length, while in Figure 3H, the analysis was based on relative blastema length. Please clarify why the difference between these parameters. Perhaps standardizing the measurement of the blastema surface area in both cases would be more accurate.

For both *eya2* mutants and Eya2 inhibition (and now in Chk1/2 inhibition) we have measured blastema length (Figure 2B and Figure 6B). For *eya2* mutants we measured zeugopod length for 30 dpa regenerates because there is no blastema present. Zeugopod was chosen because fingers are usually curved in the autopod and amputation was performed in the stylopod. So zeugopod appears to be the most unaffected portion of the limb.

4) The quantification of cell death in eya2^-/-^ is missing. The Figure 4—figure supplement 1 shows some TUNEL/ γ-H2AX staining, but it is difficult to discern what part is related to background and/or positive signal. Likewise, the amount of TUNEL + cells in the control group Figure 4I, K (below 1%) seems very low considering the DDR is activated, and regeneration requires cell death. This brings another issue related to the amount of DNA damage induced by the treatment with DMSO (Figure 4N). It seems DMSO-intact/blastema groups show a large number of cells with COMET score 3-4. If the Eya2 inhibitor is inducing DNA damage, this will account only for the slight increase of cells with COMET score four. Validating cell death with other approaches (e.g., caspase, FACS, etc.), and adding better contrast or some validation of the staining involving TUNEL may benefit the work.

The timepoints selected for the DDR analysis were specifically chosen to avoid/limit the presence of apoptotic cells observed usually during very early phases of limb regeneration. Our TUNEL data are consistent to those from previous studies. Thus, we decided that additional effort dedicated to this proposed experiment would not be the best use of time since it would likely provide minimal additional explanatory power. There were no visible TUNEL+ cells in all blastemas quantified (even though there were positive cells in the non-blastema portion of the limbs). The comet assay has background signal which most likely account for the amount of comet score 3-4 seen in the figure. All samples/data presented in each figure were acquired at the same time to subject everything to the same amount of background.

5) The model presented in Figure 5 is not intuitive, and some of the statements lack support. For example, no data is supporting the presence of replication errors. The cell cycle arrest may need additional experiments, perhaps cell cycle with FACS or other methods. Using EdU and H3P staining to deduce cell cycle arrest is probably a good start, but additional experiments are needed to support this conclusion.

We now provide a re-worked figure (now Figure 7) that we believe represent better the work performed in the paper. We now present western blot data that support cell cycle arrest at G1/S and G2/M transitions in absence of Eya2 activity.

Additionally, we provide an additional experiment that independently intersects the cell cycle. We inhibited the Checkpoint kinases 1 and 2 with a Chk1/2 inhibitor, and we found that limbs treated with this inhibitor were dramatically impaired in their regeneration.

6) The authors may want to consider expanding their characterization of factors involved in the DDR and DNA repair. The DDR influences cell fate in the presence of DNA damage, so incorporating characterization of other molecules such as p53 may strengthen the case.

We have now expanded our DDR characterization by providing more high-throughput gene expression data and qPCR validation. We have also inhibited another component of the DDR process, the regulation of the cell cycle via DNA damage checkpoint kinases Chk1/2 and found that limb regeneration is impaired.

Reviewer #3:Major critiques:1) Eyes absent homologs play roles in DNA repair, but also regulate morphogenesis of several tissues (e.g. eyes, kidneys, muscle, ears, heart) though protein-protein interactions with transcriptional regulators like Six and Dachsous family members. The fact that eya2^-/-^ does not cause increased DNA damage further complicates the authors' interpretation that DDR is the key function of Eya2. Thus, the claim that Eya2's function in axolotl limb regeneration is due to DNA damage response (and not general transcriptional dysregulation) is undersupported.a) The authors should clarify whether axolotl Eya2 plays any roles in development. The authors say that limbs are normal in eya2^-/-^ mutants, but provide only one image per genotype with no scale bars. Is the skeleton normal (e.g. zeugopod length) in uninjured limbs? The authors indicate that there is an intra-peritoneal edema in mutants, but do not discuss the etiology of this phenotype. Overall, a more comprehensive analysis of the knockout phenotype (before injury) is needed. If Eya2 plays roles in axolotl development or physiology, the authors should discuss whether other roles for eya2 might alter interpretation of their regeneration results. This might also determine whether a reader would conclude that the role of Eya2 is regeneration-specific.

We now provide a complete characterization of the *eya2^-/-^*mutants (Figure 2—figure supplement 2).

It is possible that underdeveloped kidneys, as we observed in the *eya2* mutants, could lead to a physiological difference between these animals and wild type and that these differences could indirectly impact limb regeneration. We have not found any direct precedence for this possibility in the literature, but it remains a possibility. We have added this possibility in the Conclusions.

We also leave the door open for there being a potential role for a transcription coactivation activity for Eya2 in limb regeneration (see our response to an earlier reviewer as well as our mention of this in the Conclusions).

b) The authors themselves show considerable transcriptional dysregulation after eye2 knockout (Figure 4), which further highlights other potential roles of Eya2 beyond the DDR. The markers used in Figure 4E do not seem comprehensive. Would it be worth looking at markers of muscle or neurons in the regenerating limb given other developmental roles for eya homologs?

Again, we do not rule out that there could be additional roles for Eya2 (beyond DDR) in axolotl limb regeneration. Please see previous explanations regarding this issue.

We believe that the markers are comprehensive based on the recent single-cell RNAseq papers (from our lab and the Tanaka Lab). In the single-cell data sets, *eya1* is a muscle progenitor cell marker.

We have checked neurons via histology, and we now report that in Figure 3—figure supplement 2AB. We do not observe differences in axons or in muscle in the mutants.

c) The authors should also demonstrate that perturbation of other DNA damage repair mechanisms causes similar limb regeneration failure. This should be achievable by chemical inhibition of canonical DNA damage repair machinery (e.g. inhibitors of ATM, ATR, Chk1, or Chk2).

We now provide data showing that by inhibiting Chk1/2, limb regeneration is severely impaired.

2) In several sections, the authors have quantified western blot images that are scanned from film that was developed after ECL. It is very challenging to quantify proteins accurately with this approach, due to issues around linearity of signal in both the ECL step and the development step. A great deal of caution should be exercised with these approaches and the conclusions drawn from them. If the authors wish to make quantitative statements about these western blots, more controls need to be completed to ensure that blots were in the linear range and that loading was consistent. Alternatively, another approach (e.g. fluorescent secondary antibodies rather than chemiluminescence) should be considered. Overloading or saturation looks especially likely in Figure 3A and Figure 4L which are key pieces of data for the authors' argument that Eya2 is important for DDR.

We agree that quantifying western blot images can be problematic because it relies on all the proteins being within the linear range for concentration and exposure time. ECL was used because it provides better dynamic range than fluorescent methods (especially when we blotted for cell cycle regulators). Figure 3A (now Figure 4A) clearly shows that γ-H2AX is elevated in *eya2^-/-^*blastemas and if saturation exists, it would normalize the effect rather than elevate it in this case. Unfortunately, due to sample limitations, we were not able to redo these western blots and we opted to extract protein samples from the new batch of *eya2^-/-^*animals obtained.

We did, however, try to satisfy the reviewer’s concern by using a capillary-based method for separating and quantifying protein (Protein Simple Wes system). After several trial runs, we determined that there is insufficient protein in the samples from these small blastemas for this method to be effective.

These data that appear on western blots have been corroborated with immunohistochemistry on tissue sections, and both types of analyses implicate Eya2 in DDR.

3) Validation of the H2AX antibody (and phosphospecific antibodies) seems important for the arguments put forward in this manuscript. It would be helpful to show that the authors see an increase in H2AX phosphorylation (with their antibodies and western blotting/immunostaining) after UV exposure in axolotls. This would validate the antibodies for use in this species and confirm that H2AX phosphorylation behaves as expected in axolotls. The authors should also demonstrate that the γ-H2AX staining (Figure 3E, Figure 4—figure supplement 1A?) reveals nuclear localization, as would be expected for a histone. In some images, the staining looks quite clearly cytoplasmic, though the images are not very high magnification.

We provide antibody validation in Figure 1—figure supplement 1.

Please see our previous explanation about the γ-H2AX cytoplasmic staining in the skin. We think this staining is real and that the intensity differs between wild type and *eya2* mutants. We now present a high-magnification view of this data (γ-H2AX in skin) in Figure 4—figure supplement 1A’.

Reviewer #4:Although this manuscript presents new knowledge with the first demonstration of a DNA damage response (DDR) during regeneration, I am not sure if this alone is enough to justify publication in eLife. I admit that having read the manuscript several times, I find it difficult to glean what is known and what is unknown about DDR and Eya2 in other systems. Thus, the readability and focus of the manuscript are not helping in its evaluation. From what I gather from other systems, DDR is a common feature of rapidly proliferating cells and can be a consequence of not only acute genotoxic insults but also regular replicative stress that accompanies a rapid S phase. This may be particularly true in axolotls, which must replicate over 30Gb of genome and yet can still accomplish a 48-72hour cell cycle during regeneration. If this is true, I do not find the results to highlight any regeneration-specific insights or elucidate a pressing question concerning how regeneration occurs. For me, there are two potential follow up questions that would address this.The authors do not report any effect of Eya2 mutation on the development of the organism. First, is it slower than the development of the wildtype animal? Do the authors also observe the same H2AX foci observed during regeneration? If they could show that the DDR process they observe and Eya2's specific role is unique to regeneration and does not have any role in the rapid proliferation and development of the embryo, then I think this would be quite interesting and would potentially be a mechanism where regeneration and development diverge.

We now provide a more comprehensive characterization of *eya2*^-/-^ developmental phenotype. From all aspects we have measured, including limb attributes, they are normal (see Figure 3—figure supplement 2) except for that they have underdeveloped kidneys. We are unable to count γ-H2AX foci on embryos at the moment mainly due to the limited number of mutant animals available and the nature of breeding scheme that yields a low number of *eya2*^-/-^ (Crossing of two F0 mutant animals). We agree that this could be something interesting to investigate in the future.

Secondly, the mechanism of Eya2 seems to be in the regulation of the DDR such that Eya2's phosphatase activity removes cell cycle checkpoints and indicates a successful resolution of the DNA damage. This seems to be a known role of Eya2 in other animal systems. Thus, although the authors observe phenotypes on cell cycle progression and regeneration, all DNA damage has been successfully repaired (as indicated by a normal comet assay). I think it would be more interesting to determine what happens when DNA damage cannot be repaired. Are there gross effects when progenitors accumulate unrepaired damage sites? Because regeneration can occur multiple times, does accumulated DNA damage impact subsequent regeneration events? This might be achieved through pharmacological inhibition of ATM which is the kinase that phosphorylates H2AX and marks DNA damage sites. Again, this would be unique to regeneration and be of interest to the regeneration field as well as the general stem cell field to look at the consequences of genotoxic stress on progenitor exhaustion.

The reviewer raises many interesting aspects of how DNA damage could affect

regeneration. However, our current study tries to dissect the DNA damage response

events occurring naturally during regeneration without challenging the process

externally (i.e., inducing DNA damage with UV or other means. It is known that gamma

irradiation blocks regeneration—for example, Butler 1938). For instance, when we use an Eya2 inhibitor, DNA damage is observed, which is accompanied by apoptosis and slower regeneration rates. To strengthen the DNA damage response aspect in relation to tissue regeneration, we pharmacologically inhibited Chk1/2, which are kinases playing roles in mediating the G1/S and G2/M transitions. We found that limb regeneration is severely impaired with Chk1/2 inhibition. Interestingly, G1/S and G2/M transitions were also found to be impaired in our *eya2*^-/-^ blastemas.

In terms of DNA damage in progenitor cells: our study indicates that blastema cells repair their DNA during regeneration and that cells entering the cell cycle have increased levels of γ-H2AX. There are no data supporting that DNA damage is accumulated during regeneration or how is this altered in subsequent regeneration events. These studies, while they would be quite interesting, are beyond the scope of our current study. Our goal here was test whether DNA damage response is important for regeneration, and, to accomplish that, we challenged the process by removing one of its components and analyzed the effects after a single regenerative episode. We feel that at 7 figures, 11 supplementary figures, and additional supplementary data files, embarking on this question within this work is not feasible. However, we look forward to it being addressed in the future.